# Electric-field-driven non-volatile multi-state switching of individual skyrmions in a multiferroic heterostructure

Yadong Wang[1,2], Lei Wang[3], Jing Xia[4], Zhengxun Lai[5], Guo Tian[1,2], Xichao Zhang[4], Zhipeng Hou [1,2✉], Xingsen Gao [1,2✉], Wenbo Mi [5], Chun Feng[3✉], Min Zeng[1,2], Guofu Zhou[1,2], Guanghua Yu[3], Guangheng Wu[6], Yan Zhou [4], Wenhong Wang[6], Xi-xiang Zhang [7] & Junming Liu [8]

Electrical manipulation of skyrmions attracts considerable attention for its rich physics and promising applications. To date, such a manipulation is realized mainly via spin-polarized current based on spin-transfer torque or spin–orbital torque effect. However, this scheme is energy consuming and may produce massive Joule heating. To reduce energy dissipation and risk of heightened temperatures of skyrmion-based devices, an effective solution is to use electric field instead of current as stimulus. Here, we realize an electric-field manipulation of skyrmions in a nanostructured ferromagnetic/ferroelectrical heterostructure at room temperature via an inverse magneto-mechanical effect. Intriguingly, such a manipulation is non-volatile and exhibits a multistate feature. Numerical simulations indicate that the electric-field manipulation of skyrmions originates from strain-mediated modification of effective magnetic anisotropy and Dzyaloshinskii–Moriya interaction. Our results open a direction for constructing low-energy-dissipation, non-volatile, and multistate skyrmion-based spintronic devices.

---

[1] Guangdong Provincial Key Laboratory of Optical Information Materials and Technology & Institute for Advanced Materials, South China Academy of Advanced Optoelectronics, South China Normal University, Guangzhou 510006, China. [2] National Center for International Research on Green Optoelectronics, South China Normal University, Guangzhou 510006, China. [3] School of Materials Science and Engineering, University of Science and Technology Beijing, Beijing 100083, China. [4] School of Science and Engineering, The Chinese University of Hong Kong, Shenzhen, Guangdong 518172, China. [5] Colleage of Science, Tianjin University, Tianjin 300392, China. [6] Beijing National Laboratory for Condensed Matter Physics, Institute of Physics, Chinese Academy of Sciences, Beijing 100190, China. [7] Physical Science and Engineering Division, King Abdullah University of Science and Technology, Thuwal 23955-6900, Saudi Arabia. [8] Laboratory of Solid State Microstructures and Innovation Center of Advanced Microstructures, Nanjing University, Nanjing 211102, China. ✉email: houzp@m.scnu.edu.cn; xingsengao@scnu.edu.cn; fengchun@ustb.edu.cn

Magnetic skyrmions, which are topologically nontrivial swirling spin configurations, have received increasing interest from the research community in view of their magneto-electric properties[1–25], such as topological Hall effect[24], skyrmion Hall effect[25], and ultralow threshold for current-driven motion[15–19]. These magneto-electronic properties, in combination with the nanoscale size and stable particle-like features, make magnetic skyrmions promising candidates for carrying information in future magnetic memories or logic circuits[1–4].

As the information bits, magnetic skyrmions are required to be controllably manipulated through a purely electrical manner for easy integration into modern electronic technology. To date, electrical manipulation of skyrmions has been generally realized via the use of the spin-polarized current on basis of the spin-transfer torque or spin–orbital torque effect[6,8–11,14,19]. However, in those studies, the required current density is immense, which leads to a high energy dissipation. Moreover, the Joule heating produced by the injected current is detrimental to the stability of skyrmion bits. In contrast, the electric-field (EF) method provides a potentially effective route to achieve the low-energy-dissipation and low-Joule-heating target, as the operations generate almost no current[21,22,26–33]. Moreover, the use of EF scheme can avoid the unexpected displacement of skyrmions during the writing process[26]. These features are of great significance to practical applications and have thus promoted the usage of electric field instead of current to manipulate skyrmions[21,22,26–33].

Despite recognition of the potential for the EF manipulation of skyrmions, experimental realizations are limited[21,22,26–29], especially at the room temperature[27–29]. Room-temperature EF-induced switching of skyrmions was first experimentally realized in the ferromagnet/oxide heterostructures, where the external electric field was directly applied at the ferromagnet/oxide interface to induce an interfacial charge redistribution. Consequently, a reliable binary conversation between skyrmions and ferromagnetic states was obtained[27,28]. Recently, Ma et al. have reported that such a room-temperature manipulation of skyrmions could also be realized by utilizing the magnetic-anisotropy gradient[29]. However, these methods have been demonstrated to be volatile, which may limit their further applications in spintronic devices.

In addition to the methods based on the pure ferromagnetic material systems, some recent theoretical reports have proposed the EF manipulation of skyrmions through the use of a strain-mediated ferromagnetic/ferroelectric (FM/FE) multiferroic

heterostructure[30,31]. A large in-plane strain generated from the reversal of FE polarization in the FE substrate is expected to significantly modify the interfacial magnetism of the FM layer. Thus, a reliable and controllable transition between skyrmions and other magnetic states may be envisioned. Furthermore, the strain associated with the FE polarization is non-volatile, which makes the manipulation of skyrmions non-volatile too. This feature is especially useful and essential to the design of skyrmion-based spintronic devices. Therefore, researchers are stimulated to explore approaches to realize the EF manipulation of skyrmions on basis of the FM/FE heterostructure[30,31]. Yet the experimental realization has not been implemented successfully.

In this work, we have experimentally constructed the strain-mediated FM/FE multiferroic heterostructure by combining the skyrmion-hosting multilayered stacks [Pt/Co/Ta]$_n$ ($n$ is the repetition number) with the FE substrate (001)-oriented single-crystalline 0.7PbMg$_{1/3}$Nb$_{2/3}$O$_3$-0.3PbTiO$_3$ (PMN-PT) to explore an EF manipulation of skyrmions at the room temperature. Since the design of skyrmion-based spintronic devices usually requires controllable manipulation of individual skyrmions at a custom-confined position[26,30–33], the FM layer [Pt/Co/Ta]$_n$ was fabricated into geometrically confined nano-dots with the expectation that each nano-dot could host a single skyrmion. We show that a reliable EF-induced stripe–skyrmion–vortex multistate switching can be realized, which is directly detected by the in-situ magnetic force microscope (MFM) technique. More intriguingly, such a switching is non-volatile and does not need an external magnetic field except a low one of less than 10 mT from the MFM tip.

## Results

**Fabrication of nanostructured FM/FE heterostructure.** For the multiferroic heterostructure, we opt for the (001)-oriented single-crystalline PMN-PT as the FE substrate in the view of its large piezoelectric coefficients and non-volatile strain. Meanwhile, the size of the ferroelectric domains in the (001)-oriented single-crystallized PMN-PT is ~10 μm[34], which allows us to manipulate the FM domain in a relatively large area. The multilayered stack [Pt/Co/Ta]$_n$ is selected as the target FM layer because it can stabilize sub-100 nm Néel-type skyrmions at the room temperature as a result of the balance between a large DMI, magnetic effective anisotropy, and magnetic exchange coupling[8,35,36].

Figure 1a presents the detailed structure of a typical [Pt/Co/Ta]$_{12}$/PMN-PT multiferroic heterostructure. First, magnetron sputtering is employed to deposit the multilayered stack

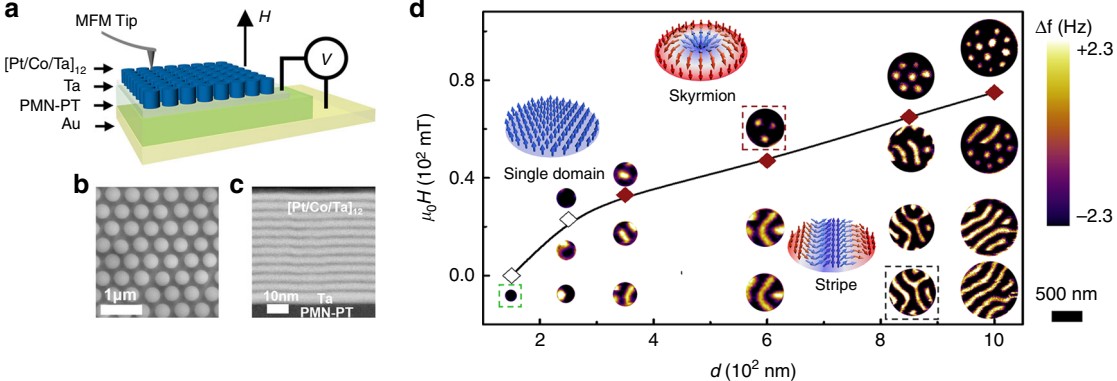

**Fig. 1 Nanostructured FM/FE multiferroic heterostructure. a** Scheme of the nanostructured FM/FE multiferroic heterostructure. **b** SEM image of ordered [Pt/Co/Ta]$_{12}$ multilayer nano-dot arrays. **c** STEM image of the cross-sections. **d** The magnetic domain evolution process in the [Pt/Co/Ta]$_{12}$ nano-dot as a function of both the external magnetic field μ$_0$H and d. The inset shows the spin textures of the magnetic domain enclosed by the color boxes. The magnetic domain enclosed by black, red, and green boxes represents the stripe, skyrmion, and single domain, respectively. The red-filled rhombuses represent the critical field where the stripe domains completely transform into skyrmions. The hollow rhombuses represent the magnetic field where the out-of-plane single domain appears. The MFM contrast represents the MFM tip resonant frequency shift (Δf). The scale bar in **c** represents 500 nm.

[Pt(2.5 nm)/Co(2.2 nm)/Ta(1.9 nm)]$_{12}$/Ta(5 nm) on the (001)-oriented single-crystalline PMN-PT. Subsequently, a two-step nano-patterning method is utilized to fabricate the [Pt/Co/Ta]$_{12}$ stack into nano-dots with diameters ($d$) ranging from 1 μm to 150 nm. More details about the fabrication processes are presented in Supplementary Fig. 1. Figure 1b provides a top-view of the heterostructure imaged with a scanning electron microscope (SEM) and demonstrates an ordered arrangement of nano-dots. By further using scanning transmission electron microscopy (STEM) to visualize its cross section (see Fig. 1c), we find that the heterostructure possess reasonably sharp interfaces between PMN-PT, Ta, and [Pt/Co/Ta]$_{12}$, which confirms the as-required structure.

To establish the $d$ range for hosting a single skyrmion, we first use MFM under different magnetic field ($\mu_0H$) to image the domain structure of the [Pt/Co/Ta]$_{12}$ nano-dots with different values of $d$ (see Fig. 1d and Supplementary Fig. 2). Notably, $\mu_0H$ represents the external magnetic field that is nominally applied to the nano-dots; however, it does not reflect the effective magnetic field, as the MFM tip also has a certain magnetic field. To minimize the influence of the magnetic field from the MFM tip on the magnetic domain structure during scanning, we select a low-moment magnetic tip for the MFM measurements (the magnetic field of MFM tip applied on the nano-dots is less than 10 mT). As illustrated in Fig. 1d, the nucleation of the skyrmions depends drastically on $d$ of the nano-dots. Both the critical magnetic field for the nucleation of skyrmions ($\mu_0H_c$) and the maximum number of skyrmions in the nano-dot ($N$) decrease correspondingly with the reduction of $d$. In particular, when $d$ is equal to 350 nm, a single skyrmion forms at a relatively low $\mu_0H_c$ of 25 mT. By further reducing $d$, no skyrmion is observed anymore while an out-of-plane single domain starts to appear in the nano-dot.

**Electric-field-induced switching of individual skyrmions.** Having established that the $d \sim 350$ nm nano-dot hosts a single skyrmion, we adopt it as the platform for exploring the EF manipulation of individual skyrmions. As indicated in Fig. 1a, when the external electric field ($E$) is applied to the heterostructure through the top Ta layer and bottom Au electrode, an in-plane strain is generated at the PMN-PT substrate. The in-plane strain is subsequently transferred to the FM nano-dots as a result of the strong mechanical coupling between nano-dots and substrate and we expect that the magnetic states in the nano-dots would be altered via an inverse magneto-mechanical effect. Since the transferred strain ($\varepsilon$) on the $d \sim 350$ nm nano-dot is difficult to be measured, the corresponding $\varepsilon$ distribution is simulated by using the finite element analysis. Details about the simulations are presented in Supplementary Fig. 3, Supplementary Note 1 and Supplementary Table 1. The simulated results demonstrate that the $\varepsilon$ distribution on the $d \sim 350$ nm nano-dot is rather inhomogeneous and relaxes along both its thickness and diameter directions. To describe the electrical field ($E$)-dependent inhomogeneous $\varepsilon$, average strain ($\varepsilon_{ave}$), which represents an overall result of the inhomogeneous $\varepsilon$, is introduced. We have derived the $E$-$\varepsilon_{ave}$ curve on basis of the relationship between simulated $\varepsilon$ distribution and $E$-dependent strain generated at the PMN-PT substrate ($\varepsilon_{sub}$). Details about the establishing processes are presented in Supplementary Fig. 4. Figure 2a summarizes $\varepsilon_{ave}$ and the corresponding images of the magnetic domain structure in a nano-dot captured by the in-situ MFM in a cycle of sweeping $E$ ranging from +10 to −10 kV cm$^{-1}$ (positive $E$ represents the direction of applied $E$ is opposite to that applied to polarize PMN-PT to saturation at initial state and negative $E$ represents the direction of applied $E$ is same as that applied to polarize

PMN-PT to saturation). At the initial state ($E = 0$ kV cm$^{-1}$, $\varepsilon_{ave} = 0\%$, and $\mu_0H = 0$ mT), only the stripe domain is observed, which suggests that the magnetic field from the MFM tip is not sufficiently large to induce the nucleation of skyrmion. As $E$ increases from 0 to +3 kV cm$^{-1}$, a tensile-type strain ($\varepsilon_{ave} > 0$) induces the stripe domain to gradually convert into a single skyrmion. Notably, in such an EF-induced transition process, no external magnetic field (excluding the magnetic field from MFM tip) is applied. Meanwhile, without strain, a magnetic field of 25 mT (excluding the magnetic field from MFM tip) has to be applied to induce the nucleation of skyrmion. This observation clearly evidences that the tensile strain has a similar function to the external magnetic field that can significantly lower the energy barrier between the stripe domain and the skyrmion and shift the skyrmion to the local energy minimum state. As we further increase $E$, the transferred strain gradually changes to a compressive type ($\varepsilon_{ave} < 0$). In contrast to the assisting effect of the tensile strain on the nucleation of skyrmion, the compressive strain promotes the skyrmion to gradually switch back to the stripe domain, indicating that the stripe domain has lower energy compared with the skyrmion under the compressive strain. When $E$ is decreased from +10 to 0 kV cm$^{-1}$, the stripe domain is restored. By sweeping $E$ toward the negative range, the variation of magnetic state exhibits a nearly symmetric behavior though the strength of both tensile and compressive strain is lower than that at the positive $E$ range. Notably, the skyrmion formed at $E = -3$ kV cm$^{-1}$ appears to be elongated, which can be ascribed to the inadequacy of the tensile strain to form a perfect skyrmion. More details about the EF-induced magnetic domain evolution process are presented in Supplementary Fig. 5. Notably, the morphology of the skyrmions is little affected by varying the tip-sample distance or reversing the magnetization of the tip (see Supplementary Fig. 6). This feature clearly suggests that the stabilization of skyrmions is little affected by the tip stray field. We also realize such a binary switching between the skyrmion and stripe domain in the [Pt/Co/Ta]$_{12}$ nano-dots with different diameters ($d \sim 600$, 850 nm) as well as in the continuous thin film, though the corresponding number of skyrmions formed in the nano-dots and the assisting magnetic fields required for the EF-induced formation of skyrmions are different (see Supplementary Fig. 7). These results strongly demonstrate that the observed EF-induced switching of skyrmions in the multiferroic heterostructure is reliable and a general phenomenon.

To investigate the EF-induced magnetic domain evolution process under larger strains, the repetition number of the FM [Pt/Co/Ta]$_n$ multilayer is decreased from 12 to 8. As expected, $\varepsilon_{ave}$ increases drastically (see Fig. 2b). Subsequently, we have examined the effects of the transferred strain on the formation of skyrmions in the $d \sim 350$ nm nano-dot. With increasing $E$ from 0 to +3 kV cm$^{-1}$, the tensile strain can also induce a stripe–skyrmion conversion, which resembles that observed in the [Pt/Co/Ta]$_{12}$ nano-dots. However, when the tensile strain becomes a compressive one with the further increase of $E$, the domain evolution process changes. We find that the compressive strain no longer makes the skyrmion return back to the stripe domain anymore but forces its spins to gradually align with the in-plane direction to form an in-plane domain at $E = +10$ kV cm$^{-1}$ (the contrast of MFM image gradually became weak). Based on previous literatures[37,38] and further analysis of the spin configuration of the in-plane domain, we confirm it as the vortex state (see Supplementary Fig. 8). After decreasing $E$ from +10 to 0 kV cm$^{-1}$, the vortex state is retained. With increasing $E$ toward the negative numerical range, the strength of both the tensile and compressive strain decreases significantly. Although the tensile strain stimulates the skyrmion to appear again, the compressive strain cannot induce the skyrmion to transform into the vortex but the stripe

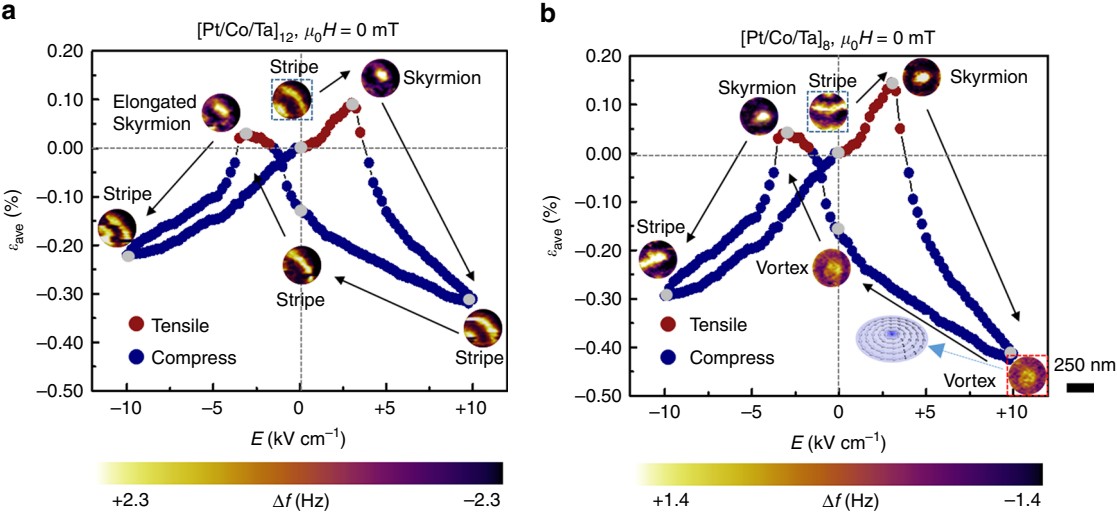

**Fig. 2 Electric-field-induced switching of individual skyrmion.** The transferred average strain $\varepsilon_{ave}$ and corresponding magnetic domain evolution processes in the $d \sim 350$ nm **a** [Pt/Co/Ta]$_{12}$ and **b** [Pt/Co/Ta]$_8$ nano-dots in a cycle of $E$ ranging from +10 to −10 kV cm$^{-1}$. Positive $\varepsilon_{ave}$ (red dots) represents tensile strain while negative $\varepsilon_{ave}$ (blue dots) represents compressive strain. $\mu_0 H$ represents the external magnetic field except that from the MFM tip and here $\mu_0 H$ is equal to be 0 mT. The inset of **b** illustrates the spin texture of the magnetic domain that is encompassed by the red box. The stripe domain enclosed by the black box shows the initial state of the magnetic domain evolution path. The gray dots represent the corresponding electric field for the MFM images. The MFM contrast represents the MFM tip resonant frequency shift ($\Delta f$). The scale bar represents 250 nm.

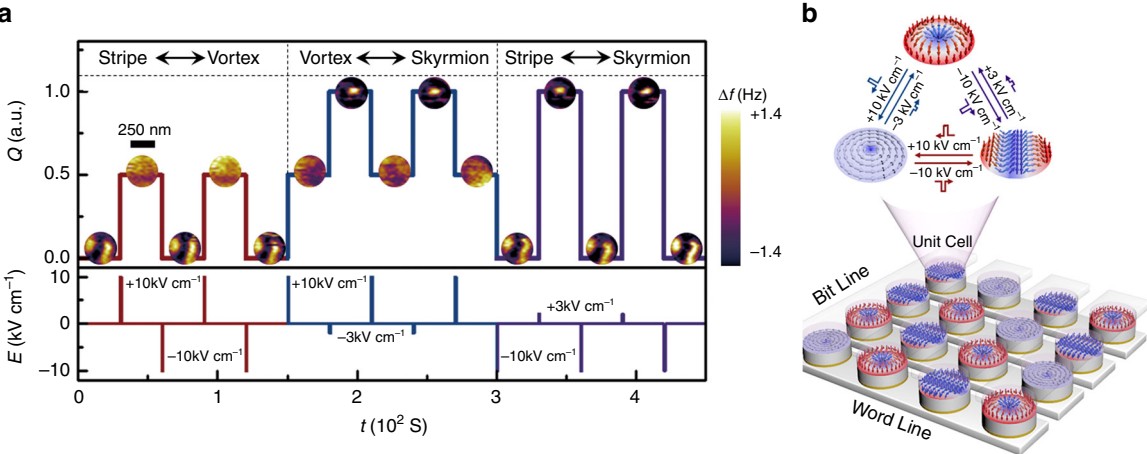

**Fig. 3 Switching of individual skyrmions induced by pulse electric field. a** Switching of topological number $Q$ of various magnetic domains ($Q = 1.0$, 0.5, and 0 corresponds to skyrmion, vortex, and stripe, respectively) by applying a pulse electric field with a pulse width of 1 ms. The insets contain the corresponding MFM images for the switching. The values of $E$ for the generation of the skyrmion, vortex, and stripe are ±3, +10, and −10 kV cm$^{-1}$, respectively. The MFM contrast represents the MFM tip resonant frequency shift ($\Delta f$). The scale bar represents 250 nm. **b** Schematic of the envisioned cross-bar random access memory device based on the FM/FE multiferroic heterostructure nano-dots with the stripe, skyrmion and vortex as storage bits.

domain. This feature suggests that the energy difference between the skyrmion and vortex exceeds that between the skyrmion and stripe domain. As a result, the compressive strain that is generated at the negative $E$ range is not sufficiently large to overcome such an energy difference to induce the formation of the vortex state. More information about the EF-induced magnetic domain evolution process and the repetition of the multistate conversion in different samples are presented in Supplementary Figs. 9 and 10.

**Non-volatile switching of different magnetic structures.** We observe that the skyrmion, stripe domain, and vortex, can be reliably switched with each other through the use of 1 ms pulses of $E = \pm3$, +10, and −10 kV cm$^{-1}$, respectively (see Fig. 3a and Supplementary Fig. 11). This finding reflects the non-volatility of the three magnetic states. We propose that such non-volatile

switching is closely related to both the large remnant strain and the geometrically confined effect.

For the non-volatility of skyrmions, the remnant strain is not the main reason, as it is nearly 0% when $E$ is reduced from ±3 to 0 kV cm$^{-1}$ (see Supplementary Fig. 12). We propose that the strong pinning effect that results from the geometrical edge of the nano-dots is essential for the stabilization of the skyrmions after an $E = \pm3$ kV cm$^{-1}$ pulse. To prove this point, we first induce the stripe domains to completely transform into skyrmions by increasing magnetic field, and subsequently the magnetic field is decreased to zero. As shown in Supplementary Fig. 12, most of the skyrmions are retained at zero magnetic field though their sharp appears to be elongated. This result demonstrates that the magnetic hysteretic effect results in the non-volatile feature of skyrmions. We have also carried out the same measurements on a continuous thin film (see Supplementary Fig. 12). However, few

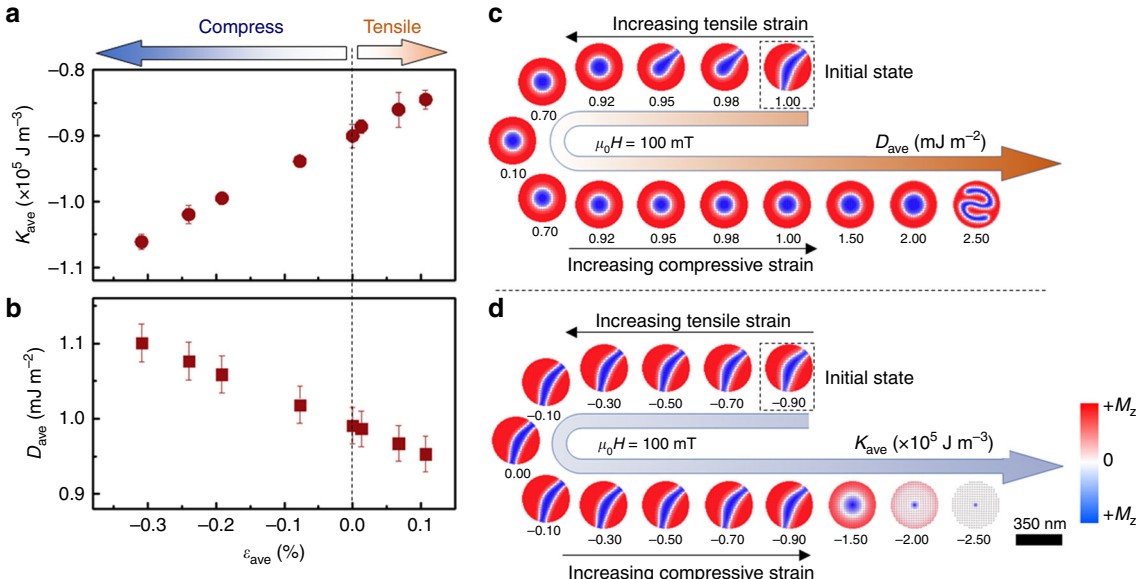

**Fig. 4 Simulated variation of $D_{ave}$ and $K_{ave}$ on magnetic domain evolution.** Dependence of the experimentally established values of **a** $K_{ave}$ (red circles) and **b** $D_{ave}$ (red squares) on $\varepsilon_{ave}$ on the $d \sim 350$ nm [Pt/Co/Ta]$_{12}$ nano-dot. The positive value of $\varepsilon_{ave}$ represents the tensile strain, while the negative value signifies the compressive strain. The dashed line represents the boundary between the tensile strain and the compressive strain. The black lines in **a** and **b** represent the fitting lines by using linear equations. The error margin of $K_{ave}$ at different $E$ is added by measuring two different samples and the error margin of $D_{ave}$ at different $E$ is added by fitting different $\varepsilon_{ave}$–$D_{ave}$ curves for both the continuous thin film and $d \sim 850$ nm nano-dot. **c** Simulations of the influence of $D_{ave}$ on the magnetic domain evolution. **d** Simulations of the variation of $K_{ave}$ on the magnetic domain evolution. Notably, when one magnetic parameter varies in the simulations, the other magnetic parameters are fixed. An external magnetic field of 100 mT is applied in the simulations, and the magnetic domain enclosed by the black dashed boxes in **c**, **d** represent the initial states in the domain evolution process. The magnetization along the z-axis ($M_z$) is represented by regions in red ($+M_z$) and blue ($-M_z$). The scale bar is 350 nm.

skyrmions are retained at zero magnetic field. This feature further suggests that the magnetic hysteretic effect in our experiments mainly originates from the geometrical pinning effect of the nano-dots, as reported in the previous literatures[9,39,40]. For the non-volatility of vortex, the case is different. As illustrated in Fig. 2b, when $E$ decreases from $+10$ to $0\,\mathrm{kV\,cm^{-1}}$, a large remnant compressive strain is obtained as a result of the hysteretic characteristic of the PMN-PT substrate. Such a remnant strain is proposed to be the main reason for stabilizing the vortex state at $E = 0\,\mathrm{kV\,cm^{-1}}$ after an $E = +10\,\mathrm{kV\,cm^{-1}}$ pulse. When we increase $E$ across $0\,\mathrm{kV\,cm^{-1}}$ toward the negative range, the strength of compressive strain decreases correspondingly and the vortex gradually transforms into the skyrmion. This feature suggests that a large enough compressive strain is essential for the stabilization of vortex. On the other hand, we find that the vortex state at $E = 0\,\mathrm{kV\,cm^{-1}}$ can reform in the nano-dot even if it is destroyed by applying an external out-of-plane magnetic field (see Supplementary Fig. 13). If the magnetic hysteretic is the key factor for the stabilization of vortex at $E = 0\,\mathrm{kV\,cm^{-1}}$, the out-of-plane single domain would transform into the stripe domain or skyrmions after decreasing the external magnetic field to zero (see Supplementary Fig. 13). This feature further confirms that the dominated factor for the stabilization of vortex at $E = 0\,\mathrm{kV\,cm^{-1}}$ is not the magnetic hysteretic effect but the large remanent strain from the ferroelectric substrate. As for the stripe domain, it is natural for us to observe that the stripe domain is reserved after an $E$ pulse of $-10\,\mathrm{kV\,cm^{-1}}$, as it is the ground state in the switching process.

In terms of practical applications, the EF-induced non-volatile, multistate switching of skyrmions is highly suitable for constructing low-energy-dissipation skyrmion-based spintronic devices, such as EF-controlled skyrmion random access memory. Figure 3b presents the schematic of an envisioned cross-bar random access memory device that employs the nanostructured

FM/FE multiferroic heterostructure with the stripe, skyrmion, and vortex as storage bits. In such a device, we expect that an EF pulse would be applied on the nano-dots via a magnetic writing head (with a low-magnetic field of less than 10 mT) to implement the information writing. To read the information bits, we propose an implementation that the nanostructured heterostructure is combined with the magnetic tunnel junctions, whose magnetoresistance (MR) can exhibit a significant variation with the switching of the stripe, skyrmion, and vortex, according to the previous reports[41].

## Discussion

We have demonstrated that a reliable EF-induced switching of individual skyrmions can be realized at the room temperature on basis of the nanostructured FM/FE heterostructure. Next, we will discuss the physics origin underlying the experimental observations. In our experiments, the EF manipulation of skyrmions originates from the strain-mediated modification of interfacial magnetism of the FM layer, such as the effective magnetic anisotropy and DMI. Hence, it is essential to explore the impact of the strain on the effective magnetic anisotropy constant ($K_{eff}$) and DMI constant ($D$). Although the magnetic exchange interaction is crucial for magnetic domain evolution as well, we propose that it is slightly influenced by the external electric field in the heterostructure, as the strain transferred to the nano-dots (maximum strain of 0.42%) is too small to induce a significant variation of magnetic exchange constant ($A$)[42,43]. More discussions about this point are shown in Supplementary Fig. 14 and Supplementary Note 2.

Based on the experimentally established relationship between $E$ and $\varepsilon_{ave}$ (see Fig. 2), we can obtain $\varepsilon_{ave}$-dependent $K_{eff}$ by measuring the $E$-dependent magnetization curves of $d \sim 350$ nm [Pt/Co/Ta]$_{12}$ nano-dots (see Fig. 4a and Supplementary Fig. 15). As known to us, strain is closely coupled with $K_{eff}$[44,45]. Thus, the inhomogeneous strain distribution should lead to an

inhomogeneous $K_{eff}$ distribution. In this work, we use the average ($K_{ave}$) of inhomogeneous $K_{eff}$ to describe the experimentally measured $K_{eff}$ on the $d \sim 350$ nm nano-dots. At $\varepsilon_{ave} = 0$ ($E = 0$ kV cm$^{-1}$), $K_{ave}$ has a negative value of $(-0.90 \pm 0.02) \times 10^5$ J m$^{-3}$, which indicates an in-plane magnetic anisotropy (IMA). Namely, the in-plane direction of the [Pt/Co/Ta]$_{12}$ layer is more easily magnetized compared with the out-of-plane direction. With the increase of tensile strain, the absolute value of $K_{ave}$ decreases correspondingly, which suggests that the tensile strain decreases IMA. In contrast, the compressive strain induces a heightened IMA with increasing the compressive strain. The changing tendency of $K_{ave}$ with the strain is similar to the observations in many other Co-based magnetic systems[44,45], which confirms our experimental results.

The corresponding absolute value of $D_{ave}$ could be calculated from[8,27,46]

$$\sigma_{DW} = 4\sqrt{AK} - \pi|D_{ave}|, \tag{1}$$

where $\sigma_{DW}$ is the domain wall surface energy density, $A$ is the exchange constant, and $K$ is the magnetic anisotropy constant that accounts for the energy difference between the spin in the domain and that in the middle of the domain wall and here is proposed to be approximately equal to that of $K_{ave}$[8,47,48]. As $\varepsilon_{ave}$-dependent $K_{ave}$ has been obtained above, the relationship between $\varepsilon_{ave}$ and $\sigma_{DW}$ should be experimentally established to calculate $\varepsilon_{ave}$-dependent $D_{ave}$. The value of $\sigma_{DW}$ can be quantified as follows by measuring the low-magnetic-field domain period ($w$) on the basis of a domain spacing model[8,27,49]

$$\frac{\sigma_{DW}}{\mu_0 M_S^2 t} = \frac{w^2}{t^2} \sum_{odd\ n=1}^{\infty} \left(\frac{1}{(\pi n)^3}\right)[1 - (1 - 2\pi nt/w)\exp(-2\pi nt/w)], \tag{2}$$

where $t$ is the thickness of the film, $M_s$ is saturation magnetization, and $w$ is the low-field domain period ($w = w_\uparrow + w_\downarrow$, $w_\uparrow$, and $w_\downarrow$ were the up and down domain widths, respectively) that can be obtained from the MFM data. For the continuous thin film or nano-dot with a relatively large diameter ($d \geq 850$ nm), they host periodic magnetic stripe domains. Supplementary Fig. 16 elaborate on the establishment of $\varepsilon_{ave}$-dependent $w$ and $\sigma_{DW}$. However, when $d$ is smaller than 850 nm ($d < 850$ nm), the strong geometrical confinement effect induces the periodic stripe domain to transform into the non-period one (see Fig. 1c). Thus, we can no longer directly calculate $D_{ave}$ of the $d \sim 350$ nm nano-dot based on $K_{ave}$ and $\sigma_{DW}$. Instead, we may derive it on basis of the $D_{ave}$-$\varepsilon_{ave}$ relationship established on the continuous thin film and $d \sim 850$ nm nano-dot because $D$ is an intrinsic parameter that originates from the spin–orbital coupling effect of the film interface and is hence little affected by the geometrical confinement. Details about establishing $D_{ave}$ of the $d \sim 350$ nm nano-dot are presented in Supplementary Figs. 17 and 18, Supplementary Notes 3 and 4, and Supplementary Table 2. Figure 4b summarizes the derived $\varepsilon_{ave}$-dependent $D_{ave}$ of the $d \sim 350$ nm [Pt/Co/Ta]$_{12}$ nano-dots. We can find that the absolute value of $D_{ave}$ decreases with the increase of tensile strain while increases with the increase of compressive strain. Such a change tendency of $D_{ave}$ with $\varepsilon_{ave}$ agrees with both previous theoretical and experimental results[43,50–52], and can be attributed to the strain-medicated modification of the electronic structure of the [Pt/Co/Ta]$_{12}$ stack[50].

After experimentally establishing the relationship between the correlative magnetic parameters and the strain, we have performed micromagnetic simulations to clarify their respective roles in the EF-induced domain evolution process. To agree with the experiments, both the $K_{eff}$ and $D$ distributions are set to be inhomogeneous based on the simulated $\varepsilon$ distribution and the experimentally established relationship among $K_{eff}$, $D$, and $\varepsilon$.

Details about the simulations can be found in Supplementary Note 5 for micromagnetic simulation. We have firstly simulated the magnetization dynamics of the $d \sim 350$ nm nano-dot on basis of the experimental magnetic parameters of the [Pt/Co/Ta]$_{12}$ heterostructure. The simulated evolution of the magnetic domain agrees well with the experimental observations (see Supplementary Fig. 19) and thus validates our theoretical model. Subsequently, we have simulated the switching of magnetic domain with the variation of $D_{ave}$ and $K_{ave}$ (see Fig. 4c, d). The simulated stripe domain at $\mu_0 H = 100$ mT (see Supplementary Fig. 19) is used as the initial state to represent the experimental domain structure at $E = 0$ kV cm$^{-1}$ and $\varepsilon_{ave} = 0\%$. As shown in Fig. 4c, d, both the absolute values of $D_{ave}$ and $K_{ave}$ are first decreased and then increased, which corresponds to the experimentally established relationship among $K_{ave}$, $D_{ave}$, $\varepsilon_{ave}$, and $E$. Our experimental results demonstrate that the transferred strain first exhibits a tensile type and increases correspondingly with the increase of $E$ from 0 kV cm$^{-1}$. The heightened tensile strain decreases both the absolute values of $D_{ave}$ and $K_{ave}$ and induces the stripe domain to convert into skyrmion. However, the simulation results demonstrate that the nucleation of skyrmion is insensitive to the variation of $K_{ave}$ but that of $D_{ave}$. As shown in Figure 4c, a slight decrease of $D_{ave}$ from 1.0 mJ m$^{-2}$ to 0.95 mJ m$^{-2}$ can induce the stripe–skyrmion transition. This feature is well consistent with the experimental observations in the [Pt/Co/Ta]$_{12}$ heterostructure and suggests that the EF-induced formation of skyrmion originates from the tensile-strain-mediated decrease of $D_{ave}$. Meanwhile, the simulations demonstrate that the skyrmion can recover back to the stripe domain by increasing $D_{ave}$. This feature is also consist with our experimental observations that the heightened compressive strain increases the absolute value of $D_{ave}$ and induces the skyrmion to convert into stripe. On basis of these results, we hence propose that that the strain-mediated variation of $D_{ave}$ has a dominant influence on the observed EF-induced stripe–skyrmion conversion in our experiments. However, we find that the variation of $D_{ave}$ cannot induce the formation of vortex that is observed in the [Pt/Co/Ta]$_8$ heterostructure even when the value of $D_{ave}$ is drastically increased to the immense value of 2.50 mJ m$^{-2}$ or decreased to only 0.10 mJ m$^{-2}$. Nevertheless, upon reducing the value of $K_{ave}$ from $-0.90 \times 10^5$ J m$^{-3}$ to $-2.00 \times 10^5$ J m$^{-3}$, namely, increasing the in-plane effective anisotropy, the vortex can form in the nano-dot. This change tendency is well consistent with our experimental observations that the heightened compressive strain increases in-plane effective anisotropy and induces the formation of the vortex. Therefore, we propose that the primary factor for the EF-induced formation of the vortex is the strain-mediated increase of IMA.

In summary, we have accomplished both a binary and multistate EF-induced switching of individual skyrmions on the basis of a nanostructured FM/FE heterostructure at the room temperature. Furthermore, such a switching is non-volatile and does not require the assistance of an external magnetic field though the low-magnetic field of MFM tips may have a minor effect. Such features reveal an approach to constructing low-energy-dissipation, non-volatile, multistate skyrmion-based spintronic devices, such as the low-energy-dissipation skyrmion random access memory. In addition, the numerical simulations evidence that the EF-induced manipulation of skyrmions originates mainly from the strain-mediated modification of the effective magnetic anisotropy and DMI. These findings offer valuable insights into the fundamental mechanisms underlying the strain-mediated EF manipulation of skyrmions.

## Methods

**Magnetic force microscope measurements.** The MFM observations are performed with scanning probe microscopy (MFP-3D, Asylum Research). For the

measurements, a low-moment magnetic tip (PPP-LM-MFMR, Nanosensors) is selected, and the distance between the tip and sample is maintained at a constant distance of 30 nm. The VFM3 component (Asylum Research) is integrated into the MFP-3D to vary the perpendicular magnetic field.

**Establishing magnetic parameters of [Pt/Co/Ta]$_{12}$ nano-dot**. The $E$-dependent $M_s$ can be obtained by measuring the $E$-dependent magnetization curves of the continuous thin film. The value of $M_s$ is proposed to be $E$-independent and established to be $697 \pm 7$ kA m$^{-3}$. The error bar of $M_s$ is added by summarizing the values of $M_s$ under different $E$. Based on the values of $M_s$ and $w$, the values of $\sigma_{DW}$ can be calculated. The error margin of $\sigma_{DW}$ is added based on the error bar of $M_s$ and $w$. The value of $K_{ave}$ is significantly affected by the variation of $E$ and the error margin of $K_{ave}$ at different $E$ is established by measuring the two different samples. The value of $A$ is obtained by fitting the temperature-dependent saturation magnetization $M_s(T)$, with the Bloch law. We find that the variation of $E$ affects $A$ slightly. Thus, the value of $A$ is proposed to be $E$-independent and established to be $16.9 \pm 0.2$ pJ m$^{-1}$. The error bar of $A$ is added fitting the $M_s(T)$ cure within different temperature range. Based on the values of $\sigma_{DW}$, $A$, and $K_{ave}$, the absolute value of $D_{ave}$ for the continuous thin film and $d \sim 850$ nm nano-dot can be directly calculated. The error margin of $D_{ave}$ is added based on the error margin of $\sigma_{DW}$, $A$, $M_s$, and $K_{ave}$. $D_{ave}$ of the $d \sim 350$ nm nano-dot is derived it on basis of the $D_{ave}$–$\varepsilon_{ave}$ relationship established on the continuous thin film and $d \sim 850$ nm nano-dot because $D$ is an intrinsic parameter that originates from the spin–orbital coupling effect of the film interface and is hence little affected by the geometrical confinement.

**Reporting summary**. Further information on research design is available in the Nature Research Reporting Summary linked to this article.

## Data availability

All relevant data that support the plots within this paper are available from the corresponding author upon reasonable request.

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

## Acknowledgements

The authors thank for the financial supports from the National Key Research and Development Program of China (Nos. 2016YFA0201002 and 2016YFA0300101), the National Natural Science Foundation of China (Nos. 11674108, 51272078, 11574091, 51671023, 51871017, 11974298, and 61961136006), Science and Technology Planning Project of Guangdong Province (No. 2015B090927006), the Natural Science Foundation of Guangdong Province (No. 2016A030308019), Open Research Fund of Key Laboratory of Polar Materials and Devices, Ministry of Education, National Natural Science Foundation of China Youth Fund (Grant No. 51901081), Science and Technology Program of Guangzhou (No. 2019050001), President's Fund of CUHKSZ, Longgang Key Laboratory of Applied Spintronics and Shenzhen Peacock Group Plan (Grant No. KQTD20180413181702403), Guangdong Basic and Applied Basic Research Foundation (Grant No. 2019A1515110713).

## Author contributions

Z.H. and X.G. conceived and designed the experiments. C.F., L.W., and Y.W. synthesized the heterostructures. X.C.Z., J.X., and Y.Z. performed the micromagnetic simulations. The manuscript was drafted by Z.H. and X-.X.Z. with contributions from G.T., Z.L., W.M., M.Z., G.Z., G.Y., X.G., G.W., W.W., and J.L. All authors discussed the results and contributed to the manuscript.

## Competing interests

The authors declare no competing interests.
