## [Peer Review File · Nature Communications]

Reviewers' comments:

Reviewer #1 (Remarks to the Author):

The manuscript 'Electric-field-driven Non-volatile Multi-state Switching of Individual Skyrmions in a Multiferroic Heterostructure' by Wang et al. reports on a combined magnetic force microscopy (MFM) and simulations study. A heterostructure of a magnetic film on a ferroelectric film is investigated regarding the electric field dependent magnetic state. In particular nano-dots of different size and thickness are imaged by MFM. The authors identify a switching between different magnetic states upon changing the electric field applied across the ferroelectric. By comparison with simulations they aim to recover electric field dependent material parameters to unravel the underlying physics and conclude that strain in the magnetic film affects the anisotropy and the Dzyaloshinskii-Moriya interaction (DMI). The manuscript is easy to follow and all steps of the interpretation are explained and supported with additional material in the supplementary information. While the topic of electric field switching of skyrmions is of interest, I have several major concerns about the presented conclusions and thus cannot recommend publication in Nature Communications.

The major concerns are listed below.

(1) Strain

I am confused by the data presented in Fig. 2 (I am not an expert on ferroelectrics).

(1a) Do I understand correctly, that the data points for the strain are the ones measured for the continuous magnetic film (because it is difficult to measure on the nano-dots as stated in the text)?

This is then very misleading to show it like this.

Regarding the quantitative strain: compared to the film the nano-dot will relax more strongly, it will be affected to a comparable amount as the film at the interface to the FE, but relax much more rapidly as the distance to that interface increases; thus the given epsilon is an upper limit.

(1b) What breaks the symmetry of the +/-E, why is the strain not symmetric around $E = 0$? If this is the loop, what is the strain in the virgin state at $E = 0$? How can you discriminate between hysteretic effects from the FE and hysteretic effects that are due only to the magnetic material (e.g. skyrmion states have been reported at zero magnetic field after a sweep of the external magnetic field, even though they are not the zero magnetic field ground state, e.g. in Milde et al., Science 340, 1076 (2013)). A comparison of several MFM images with the same epsilon but different E and history would be necessary to possibly disentangle these effects.

(2) Non-volatile

The authors claim that their method is superior over previous results because it is non-volatile: 'the strain that is associated with the FE polarization is non-volatile, which makes such manipulation of skyrmions non-volatile too (line 99), and 'We proposed that such non-volatile switching closely related to both the large remnant strain and the geometrically confined effect' (line 220). How can you exclude that there is also here charge accumulation at the interface as a driving force? How can you disentangle from magnetic state remanence (see previous question)? (this is also connected to my general question above whether the strain is remanent and what the virgin state is).

(3) Magnetic field

Did the magnetic field from the MFM tip have an influence on the magnetic state? This could be identified by either tip-sample distance dependent measurements or measurements with opposite direction of the external magnetic field.

(4) Anisotropy

Is the way to calculate K_{eff} as explained in Fig. S10 a typical way to do this? For which kind of

systems is this method valid? As I understand there are many different magnetic phases that appear while sweeping the external magnetic field that contribute to the measured total magnetization. One can see that the hysteresis loops do not have the typical S-shape of a ferromagnet but instead reflect transitions between different magnetic phases. Is the evaluation regarding K_{eff} still valid for such a kind of more complex system?

The authors state that 'an out-of-plane single domain started to appear in the nano-dot' (line 150). Why would the single domain be out-of-plane when there is an in-plane anisotropy?

The color scale in the simulated images ranges from red over white to blue. In the majority of the presented images the color scale suggests, that the magnetic state is dominantly out-of-plane, this does not seem likely for a system with in-plane anisotropy. How is the color scale set? Is it identical for all images?

(5) DMI

The authors state that 'Based on the values of σ_{DW} , A , and K_{eff} , we derived the absolute value of D' (line 461). The σ_{DW} (domain wall energy) was derived from the domain widths as explained in Fig. S11. Here the authors use simulations with some material parameter input values to simulate MFM images which are then compared to the MFM experimental data to determine the wall width. This method is not a reasonable way to obtain these values. Thus also the derived σ_{DW} and thus also the derived DMI are not reliable. The period of the stripe domain is a measure for the DMI: for a given A and K the DMI sets the length scale; the domain wall width is independent from the DMI, only the domain wall energy depends on the DMI (then still the problem remains how to determine K and whether A can be treated as strain-independent).

(6) Vortex

This MFM-images that are interpreted as vortex states do not look like what I expect for a vortex measured with MFM, see Shinjo et al., Science 289, 930 (2000). Why do you not see the out-of-plane region in the vortex in MFM? Which material parameters do the spin textures in Fig. S6 correspond to, in particular how large is the vortex core and is that reasonable?

The authors mention an energy barrier between the skyrmion and the vortex (line 209), but why would there be an energy barrier when that is a smooth continuous transition!?

How can geometry favor a vortex state over a skyrmion state? Either there is in-plane or out-of-plane anisotropy, that should be the decisive parameter to select one of the two.

(7) Color scale

The color bar used for the experimental data is not clear, also to compare images of the same series (i.e. identical magnetic tip and lift mode height) the color scale should be set identical for all measurements to allow better comparison (or at least stated how the color scale was set).

Reviewer #2 (Remarks to the Author):

Review on NCOMMS-19-30237 : Electric-field-driven Non-volatile Multi-state Switching of Individual Skyrmions in a Multiferroic Heterostructure

Manuscript ID : NCOMMS-19-30237

Author(s): Yadong Wang et al.

Authors investigated electric-field manipulation of individual skyrmions in a nanostructured ferromagnetic/ferroelectric heterostructure at room temperature via an inverse magneto-mechanical effect. In the present research work the authors showed that such manipulation of skyrmions is non-volatile and exhibits a multi-state feature. Micromagnetic simulation results indicate that the electric-field manipulation of skyrmions originates predominantly from the strain-mediated modification of effective magnetic anisotropy and Dzyaloshinskii-Moriya interaction.

The paper is well organized and the presentation of the results is clear. The conclusions are soundly supported by data presented in the manuscript and in the supplementary data. The paper is very interesting, well written and provides concise explanations, justifications and physical insights towards new directions regarding the construction of low energy dissipation, non-volatile, and multi-state skyrmion-based spintronic devices.

I recommend this paper for publication in "Nature Communications " subject to a number of minor changes and corrections that could be addressed. In order to improve readability I suggest the following:

1. The authors should try to shorten the lengthy Introduction section (compared to the actual manuscript's presentation discussion; approximately 2 pages from a total of 11 pages). In particular, the authors should trim down lines 103-119 of page 4. The same discussion naturally belongs to the summary which is nicely given by the authors in pages 11-12 (lines 320-329).
2. Page 4 line 100: stimulated.
3. The present study and the experimental procedure took place at room temperature. The micromagnetics simulations presented in Methods Section express the energy density having contributions from exchange, anisotropy, demagnetization and Dzyaloshinskii-Moriya. No thermal field is introduced or implicitly stated. Presumably, the micromagnetic simulations took place at 0K. The authors should comment on this in a more detailed manner.
4. Page 10 line 279. The figure 4b should be corrected. The epsilon-dependent D refers to figure 4a.

Reviewer #3 (Remarks to the Author):

The article describes how the magnetic texture in nano discs is modified under strain. The strain is modulated using an electric field on a piezoelectric (also ferroelectric) PMN-PT material. The magnetization texture can switch from in-plane magnetization to out-of-plane hosting worm domain or skyrmions, depending upon electric, magnetic field, and disc diameter. These experimental results are impressive.

The English is globally good, but all verbs are in past tense, which read very strange sometimes: e.g. "The details of the simulations were provided in the Methods section". They are still in this section... The text is sometimes very lengthy describing the curves in the figures. I am shocked by the use of different types of units, sometimes even in the same sentence (ll.447-448) or figure (4): erg, J, V, Oe, mT, ... Authors should stick to modern SI units! Because of this poor consistency, the reader does not know a priori if the formula in the text are expressed in the cgs/emu or SI unit system (or even worse, a mix of them!?). Punctuation after equations should be revised. Figures are OK, except for the units or other details.

I think that this article deserves a bit more work before being published. The results are already fine and interesting, but their interpretation and description in the article could be improved substantially by a few more experiments. I suggest a list of possible improvements below.

lines 115-116 + 141-145: "except a low [field] of less than 100 Oe from the MFM tip": In fact this field is applied only when the tip flies over the surface (for topography or MFM). I do not believe that the skyrmion is written by the tip while measuring the disc. Usually, the opposite phenomenon happens: the skyrmion is deleted by the tip. Authors could check if the skyrmion changes shape when reversing the magnetization of the MFM tip! This would avoid useless (?) lengthy comments about the tip stray field.

lines 152-162: I find the description of the PMN-PT a bit incomplete. What is the orientation of the substrate? Is it a single crystal? Does the reversal occur by the domain nucleation and growth? If yes, what is the typical size of the domains compared to the discs or the skyrmions? Could it play a role?
line 179: "after turning off E" is unclear. What means "reserved"? Do you mean "restored"?

lines 216-217: The remnant ferroelectric state induces several magnetic states: skyrmions, stripe and vortex. This is the most interesting/surprising point of the article. To substantiate the discussion about the pinning, I would suggest performing small magnetic field cycles. If the magnetic states at ferroelectric remanance results from pinning, then a field cycle should bring the magnetic system to another state. It would be rather simple to verify it, and it would strengthen very much the discussion.
Eq.1, l.276: Please precise about the K_{eff} : it is the effective magnetic anisotropy of the domain wall or the film?

page 10 and 11: The magnetic parameters are indicated with extremely high precision. Can the authors give error margin to all of them. They should also discuss what is the displayed error margin. For example, in Fig.4, the error margins are just "relative", authors probably assumed a perfect knowledge of the saturation magnetization (M_s) or exchange energy (A). I doubt that they can determine M_s much better than within a few percent precision, or even few tens of % for A ...

Fig.1c: Why reversing the scale of d ? It is very confusing! Moreover, the choice of field at which skyrmions are nucleated is a bit arbitrary. Indeed from the pictures of Fig.S2, one can see that skyrmions appear at lower field than indicated, coexisting with worm domains. Authors should rather indicate a range of field at which the transition occurs.

Fig.1 caption: line 501 – 'd' not 'D'! The caption does not describe what represent the data or the line!!

line 502: the colours do not correspond to the boxes.

Fig.2 caption: line 548 – there is no box.

Response to the Report of Referee A

Referee A's General Comment: The manuscript 'Electric-field-driven Non-volatile Multi-state Switching of Individual Skyrmions in a Multiferroic Heterostructure' by Wang et al. reports on a combined magnetic force microscopy (MFM) and simulations study. A heterostructure of a magnetic film on a ferroelectric film is investigated regarding the electric field dependent magnetic state. In particular nano-dots of different size and thickness are imaged by MFM. The authors identify a switching between different magnetic states upon changing the electric field applied across the ferroelectric. By comparison with simulations they aim to recover electric field dependent material parameters to unravel the underlying physics and conclude that strain in the magnetic film affects the anisotropy and the Dzyaloshinskii-Moriya interaction (DMI). The manuscript is easy to follow and all steps of the interpretation are explained and supported with additional material in the supplementary information. While the topic of electric field switching of skyrmions is of interest, I have several major concerns about the presented conclusions and thus cannot recommend publication in Nature Communications.

Authors' Reply: We sincerely thank the reviewer for careful reading of our manuscript. The valuable suggestions and comments are greatly helpful to improve our manuscript. Following the referee's comments and suggestions, we have carried out additional experiments and analysis to improve the interpretation and description of our manuscript. Below we answer the reviewer's questions and comments in a point-by-point basis. We hope the reviewer will be satisfied with the revised manuscript and our responses.

Referee A's Comment #1: (1) Strain. I am confused by the data presented in Fig. 2 (I am not an expert on ferroelectrics). (1a) Do I understand correctly, that the data points for the strain are the ones measured for the continuous magnetic film (because it is difficult to measure on the nano-dots as stated in the text)? This is then very misleading to show it like this. Regarding the quantitative strain: compared to the film the nano-dot will relax more strongly, it will be affected to a comparable amount as the film at the interface to the FE, but relax

much more rapidly as the distance to that interface increases; thus the given epsilon is an upper limit.

Authors' Reply: We thank the reviewer for giving the valuable comments. In this work, we would like to present the E - ε curves to describe the transferred strain on the nano-dots in a quantitative manner. However, at the current stage, it is really a challenge to directly measure the strain on the nano-dot surface due to the extremely small size of the nano-dots. Instead, we assume that the transferred strain on the nano-dots is approximately equal to that on the continuous ferromagnetic (FM) film, and meanwhile the transferred strain on the continuous film is quantitatively measured by using a resistance-strain gauge (see Fig. 2 in the main text).

We agree with the reviewer that the transferred strain on the nano-dot relaxes more strongly than that on the continuous film. The strain on the nano-dot generally relaxes along its thickness (out-of-plane) and diameter (in-plane) directions because both the top surface and side surface of the nano-dot are free. With the decrease of diameter (the thickness of the nano-dot is fixed), the ratio of side surface to the total surface increases correspondingly, which makes the side surface play an increasingly important role in the relaxation of strain. Thus, the transferred strain on the nano-dot relaxes more and more strongly with the decrease of the diameter. To estimate the transferred strain on the nano-dot, we have simulated the corresponding strain distribution by using the finite element analysis (FEA). Details about the simulations are presented in Fig. R1 and Fig. R1 note. We have first simulated the strain distribution on a nano-dot with a very large diameter (d) of 2 μm and a thickness (t) of 60 nm (to represent the continuous thin film). In such a nano-dot, the relaxation of strain at the central region is mainly affected by the thickness and the relaxation effect along the diameter can be ignored due to an extremely large ratio of diameter to thickness (approximately to be 33). As shown in Fig. R1, when a biaxial compressive strain of 0.4% is applied on the substrate, the transferred strain on the central region of the top surface decreases slightly to 0.38% and the corresponding ratio of transferred strain to the strain at the substrate ($\varepsilon_{\text{top}}/\varepsilon_{\text{substrate}}$) is calculated to be 95%. This feature suggests that the strain does not relax significantly along the thickness direction in such a nano-dot ($d \sim 2 \mu\text{m}$, $t \sim 60 \text{ nm}$). With the decrease of d (the thickness t of the nano-dot is fixed to be 60 nm), the relaxation effect along

the diameter becomes increasingly pronounced and the value of $\epsilon_{\text{top}}/\epsilon_{\text{substrate}}$ decreases approximately in an exponential manner. In the case of the $d \sim 350$ nm nano-dot, the transferred strain is reduced by 50% compared with the strain applied at the substrate. When d decreases to 50 nm, the transferred strain is reduced drastically by more than one order of magnitude. These results clearly suggest that the transferred strain relaxes more and more strongly with decreasing the diameter of the nano-dot.

In the initial manuscript, we said that “the transferred strain on the nano-dots is approximately equal to that on the continuous FM thin film”. On basis of our simulations and the reviewer’s comments, we find that such a statement is inexact. In the revised manuscript, we have corrected this point and mentioned that the measured strain shown in Fig. 2 is just an upper limit (see Page 6, lines 155-160). Meanwhile, we have added new simulations in the Supplementary Information to give readers an estimated value of the transferred strain on the nano-dots (see Supplementary Fig. S4).

Figure R1 a. Finite element contour plots of strain on nano-dots with different diameters ranging from 50 nm to 2000 nm subjected to a constant biaxial compressive strain of 0.4% from the PMN-PT substrate. **b.** The dependence of $\epsilon_{\text{top}}/\epsilon_{\text{substrate}}$ on the diameter of the nano-dot. $\epsilon_{\text{top}}/\epsilon_{\text{substrate}}$ represents the ratio of transferred strain on the central region of the top surface to the strain applied at the substrate. The data is well fitted by using an exponential equation: $\epsilon_{\text{top}}/\epsilon_{\text{substrate}} = 1 - \exp(-2d)$.

Figure R1 Note: The strain distribution is simulated by using the Abaqus v6.14-1 software. The material parameters of PMN-PT used for the simulation are given in Table R1^[1]. The elastic moduli and Poisson’s constants of Ta, Co and Pt are given as 186 GPa, 215 GPa, 171 GPa, and 0.34, 0.33, 0.34, respectively^[2,3]. Here, the multi-layered nano-dot is simplified into a homogeneous structure, where the elastic parameters are approximately calculated as 190

GPa and 0.34 (see Reference [4] for the details of approximation).

References:

- [1] Zhang, R., Jiang, B. & Cao W. Elastic, piezoelectric, and dielectric properties of multidomain $0.67\text{Pb}(\text{Mg}_{1/3}\text{Nb}_{2/3})\text{O}_3\text{-}0.33\text{PbTiO}_3$ single crystals. *J. Appl. Phys.* **90**, 3471 (2001).
- [2] <https://wiki.tw.wjtk.site/wiki/Tantalum>.
- [3] <https://www.azom.com/properties.aspx?ArticleID=596>.
- [4] <http://web.mit.edu/course/3/3.11/www/modules/laminates.pdf>.

$s_{11}^E(\text{m}^2\text{N}^{-1})$	$s_{33}^E(\text{m}^2\text{N}^{-1})$	$s_{12}^E(\text{m}^2\text{N}^{-1})$	$s_{13}^E(\text{m}^2\text{N}^{-1})$	$s_{44}^E(\text{m}^2\text{N}^{-1})$
69.0	119.6	-11.1	-55.7	14.5
$s_{66}^E(\text{m}^2\text{N}^{-1})$	$d_{33}^E(\text{C}\text{N}^{-1})$	$d_{31}^E(\text{C}\text{N}^{-1})$	ϵ_{33}/ϵ_0	
15.2	-1338	2820	6650	

Table R1 The elastic and piezoelectric parameters of PMN-PT [1]. ϵ_0 denotes the dielectric constant of air.

Referee A’s Comment #2: (1b) What breaks the symmetry of the +/-E, why is the strain not symmetric around $E = 0$? If this is the loop, what is the strain in the virgin state at $E = 0$?

Authors’ Reply: We thank the reviewer for careful reading our manuscript. A ferroelectric is an insulating system that possesses two or more discrete stable or metastable states with nonzero electric polarization (P) in zero electrical field (E), referred to as “spontaneous polarization”[1]. When a certain electrical field is applied, these polarized states can be modified through the coupling of the electric field to the polarization ($-E \cdot P$) [1]. Moreover, the crystal structure of ferroelectric is closely coupled with the polarized states, which makes that the ferroelectric crystal is generally elongated along the polarized direction[1]. On basis of the two important features, the crystal structure of the ferroelectric can be modified by the electric field. Together with such a structural modification, a strain can be further generated in the ferroelectric crystal[1].

Take the ferroelectric PMN-PT used in our experiments for example. It has a rhombohedral crystal structure with eight spontaneous polarizations along the $\langle 111 \rangle$ directions, denoted as $r1^+/r1^-$, $r2^+/r2^-$, $r3^+/r3^-$, $r4^+/r4^-$ (as shown in Fig. R2a)[2-4]. Initially, the direction of the polarized states is random, which makes the PMN-PT exhibit a non-polarized state on the whole. In our experiments, we first applied a large external electrical field of -10 kV/cm along

the [001]-direction of the PMN-PT to polarize it to saturation (see Fig. R2b). In such a saturation state, the polarized vectors are oriented along the $r1^-$, $r2^-$, $r3^-$, and $r4^-$, as shown in Fig. R2b. Meanwhile, the spontaneous polarization elongates the c -axis of the crystal lattice while the in-plane a -/ b -axis of the lattice is compressed. Hence, a compressive strain is generated at the top surface of the PMN-PT substrate and subsequently transferred to the magnetic film deposited on the PMN-PT.

Hereafter, the external electrical field is decreased to 0 kV/cm. At $E = 0$ kV/cm, the direction of the polarized vectors keeps unchanged while their strength decreases (see Fig. R2c). The decrease of the strength of polarized vectors decreases the in-plane deformation of PMN-PT crystal and hence reduces the compressive strain that is transferred on the top film. As sweeping the electric field across 0 kV/cm to the polarized reversal field of approximately +3 kV/cm, the strength of compressive strain further decreases correspondingly (see Fig. R2c-e). By referring to previous literatures^[5-7], the strain state at $E = 0$ kV/cm (decreasing to zero after a saturated polarization) is generally set as the initial state (the strain is set to be 0%, see Fig. 2 of the main text) though there still exists a certain compressive strain. Since the compressive strain at the positive electric field is smaller than that at $E = 0$ kV/cm, the strain state in the positive electric field range from 0 to +3 kV/cm can be regarded as the tensile type and the tensile strain increases with the increase of E ^[5-7] (as shown in Fig. 2 of the main text) when the strain state at $E = 0$ kV/cm is set to be 0%.

When E increases above the polarized reversal field of PMN-PT (approximately +3 kV/cm), the spontaneous polarizations are switched by 71° , 109° , and 180° ^[3]. As shown in Fig. R2e, the polarization variants changing from $r1^-/r3^-$ to $r2^+/r4^+$ or $r2^-/r4^-$ to $r1^+/r3^+$ corresponds to the 109° switching, polarization variants changing from $r1^-$ to $r3^+$ or $r3^-$ to $r1^+$ (the same case for $r2^-/r4^-$) correspond to the 71° switching, and polarization variants without any category change (e.g., $r1^-$ to $r1^+$ and so on) correspond to the 180° switching. With increasing E from polarization reversal field of +3 kV/cm to +10 kV/cm, the strength of polarized vectors gradually increases towards the opposite direction (see Fig. R2e and 2f), which increases the in-plane deformation of PMN-PT crystal. Hence, the compressive strain transferred on the magnetic film increases correspondingly with the increase of E and reaches a maximum at

+10 kV/cm. When E is decreased from +10 kV/cm to 0 kV/cm (E_{return}), a large remanent strain is observed (see Fig. R2g). Such a remanent strain is proposed to be closely related to the polarization variation of 109° switching^[3,8]. For the $71^\circ/180^\circ$ switching, the in-plane polarization component keep unchanged after E returns back to zero (see Figs. R3a and 3b). This feature suggests that in-plane deformation of the PMN-PT crystal also keeps unchanged and the corresponding remanent strain state at E_{return} should be equal to that at E_{initial} . However, in the case of the 109° rotation, the in-plane polarization component is rotated by 90° (see Fig. R3c)^[3,7]. As mentioned above, the PMN-PT crystal shows a rhombohedral structure, which makes the in-plane component of the polarized vectors at the two states (E_{initial} and E_{return}) have a large discrepancy (see Fig. R3c). This features further leads to a different in-plane deformation of the PMN-PT at E_{initial} and E_{return} and hence results in the strain asymmetry at $E = 0$ kV/cm (E_{initial} and E_{return}). With further increasing E towards the negative range, the polarization vectors that corresponds to the $71^\circ/180^\circ$ switching leads to a symmetry variation of strain compared with that at the positive E range while the polarization vectors corresponding to the 109° switching leads to an asymmetric variation of strain due to their discrepancy of the in-plane polarization component. Thus, the polarization vectors corresponding to the 109° switching breaks the symmetry of the strain at $\pm E$.

References:

- [1] Rabe, K. M., Dawber, M., Lichtensteiger, C., Ahn, C. H. & Triscone, J. M. *Modern Ferroelectrics* (Springer, 2007).
- [2] Ghidini, M. et al. Shear-strain-mediated magnetoelectric effects revealed by imaging. *Nat. Mater.* **18**, 840 (2019).
- [3] Zhang, S. et al. Electric-field control of nonvolatile magnetization in $\text{Co}_{40}\text{Fe}_{40}\text{B}_{20}/\text{Pb}(\text{Mg}_{1/3}\text{Nb}_{2/3})_{0.7}\text{Ti}_{0.3}\text{O}_3$ structure at room temperature. *Phys. Rev. Lett.* **108**, 137203 (2012).
- [4] Zhu, Q. X. et al. Ultrahigh Tunability of Room Temperature Electronic Transport and Ferromagnetism in Dilute Magnetic Semiconductor and PMN-PT Single-Crystal-Based Field Effect Transistors via Electric Charge Mediation. *Adv. Funct. Mater.* **25**, 1111-1119. (2015).
- [5] Buzzi, M. et al. Single domain spin manipulation by electric fields in strain coupled artificial multiferroic nanostructures. *Phys. Rev. Lett.* **111**, 027204 (2013).
- [6] Jiang, C., Zhang, C., Dong, C., Guo, D. & Xue, D. Electric field tuning of non-volatile three-state magnetoelectric memory in $\text{FeCo-NiFe}_2\text{O}_4/\text{Pb}(\text{Mg}_{1/3}\text{Nb}_{2/3})_{0.7}\text{Ti}_{0.3}\text{O}_3$ heterostructures. *Appl. Phys. Lett.* **106**, 122406 (2015).
- [7] Yang, L. et al. Bipolar loop-like non-volatile strain in the (001)-oriented $\text{Pb}(\text{Mg}_{1/3}\text{Nb}_{2/3})\text{O}_3\text{-PbTiO}_3$ single crystals. *Sci. Rep.* **4**, 4591 (2014).
- [8] Ba, Y. et al. Spatially Resolved Electric-Field Manipulation of Magnetism for CoFeB

Figure R2 a. Schematic of the eight possible spontaneous polarized states in a PMN-PT unit cell. Schematic of the polarized states under an external electrical field of **b** -10 kV/cm, **c** 0 kV/cm after decreasing E from -10 kV/cm, **d** +1 kV/cm, **e** +3 kV/cm, **f** +10 kV/cm, and **g** 0 kV/cm after decreasing E from +10 kV/cm. The strain state under these electrical fields is listed below the schematic.

Figure R3 Upper panel of **a** is the schematic of 71° polarization switching. Lower panel of **a** is the schematic of 71° polarization switching seen from along the c -axis. The dotted rhomboids represent the in-plane structure of PMN-PT crystal at E_{initial} and E_{return} . Upper panel of **b** is the schematic of 180° polarization switching. Lower panel of **b** is the schematic of 180° polarization switching seen from along the c -axis. Upper panel of **c** is the schematic of 109° polarization switching. Lower panel of **c** is the schematic of 109° polarization switching seen from along the c -axis.

Referee A's Comment #3: How can you discriminate between hysteretic effects from the FE and hysteretic effects that are due only to the magnetic material (e.g. skyrmion states have been reported at zero magnetic field after a sweep of the external magnetic field, even though they are not the zero magnetic field ground state, e.g. in Milde et al., *Science* 340, 1076 (2013)). A comparison of several MFM images with the same epsilon but different E and history would be necessary to possibly disentangle these effects.

Authors' Reply: We thank the reviewer for his/her valuable comments. Our experimental

results demonstrate that the electrical-field-driven switching of different magnetic states is non-volatile. We propose that both the large remnant strain of ferroelectric (FE) substrate and the magnetic hysteretic effect originating from the geometrically pinning, result in the non-volatile feature.

For the non-volatility of skyrmions, our experimental results show that the magnetic hysteretic effect originating from the geometrically pinning is the main reason. As shown in Fig. 3 in the main text, when E increases from 0 to +3 kV/cm, the tensile strain induces the stripe to gradually transform into skyrmions. After E is decreased from +3 kV/cm to 0 kV/cm, the skyrmions can be retained. Since the value of E for skyrmion generation (+3 kV/cm) is nearly the same with the polarization reversal electric field (+3 kV/cm) of the PMN-PT substrate, the remnant strain is nearly 0% when E is decreased from +3 kV/cm to 0 kV/cm (see Fig. R4a). Thus, the remnant strain of FE substrate is not the main reason for the non-volatility of skyrmions. Instead, we propose that the magnetic hysteretic effect originating from the geometrically pinning is the main reason. To prove this point, we first induce the stripe domains to completely transform into skyrmions by increasing magnetic field (see Fig. R4b), and subsequently the magnetic field is reduced to zero. As shown in Fig. R4b, most of the skyrmions are retained at zero magnetic field though their shape appears to be elongated. This result demonstrates that magnetic hysteretic effect results in the non-volatile feature of skyrmions. We have also carried out the same measurements in a continuous thin film (see Fig. R4c). However, few skyrmions are retained at zero magnetic field. This feature further suggests that the magnetic hysteretic effect in our experiments mainly originates from the geometrical pinning effect of the nano-dots, as reported in the previous literatures^[1-3]. In the literature^[4] pointed by the reviewer, the skyrmions in the continuous film can be retained at zero magnetic field after a sweep of the external magnetic field. In their work, the sweeping field process is carried out after a field cooling (FC) process. Such a FC process is highly beneficial for the stabilization of skyrmions at zero magnetic field, as demonstrated in other literatures^[4,5]. However, the sweeping field process in our experiments is directly carried out at room temperature. Thus, few skyrmions are retained at zero magnetic field.

In the case of the non-volatility of the vortex state, we propose that the large remnant strain of FE substrate plays a dominated role. Our experimental results have demonstrated that, as E increases from 0 to +10 kV/cm, the compressive strain induces the stripe domain or skyrmion to gradually transform into the vortex state. This feature suggests that the compressive strain is the key factor for the formation of vortex. After E decreases from +10 kV/cm to 0 kV/cm, the vortex can be retained. As shown in Fig. 2b in the main text and Fig. R4a, when E decreases from +10 kV/cm to 0 kV/cm, a large remnant compressive strain of approximately 0.17% can be obtained as a result of the hysteretic characteristic of the PMN-PT substrate. When we increase E across 0 kV/cm towards the negative range, the strength of compressive strain decreases correspondingly and the vortex gradually transforms into the skyrmion. This feature suggests that a large enough compressive strain is essential for the stabilization of vortex. On the other hand, we find that the vortex state at $E = 0$ kV/cm can reform in the nano-dot after it is destructed by applying an external out-of-plane magnetic field. Fig. R5a shows a typical MFM image at $E = 0$ kV/cm after E decreases from +10 kV/cm. We can find that the nano-dots host vortex states at $E = 0$ kV/cm. By increasing the magnetic field from 0 mT to 50 mT, the vortex states gradually transform into the out-of-plane single domain. When the magnetic field is decreased to 0 mT, the vortex domains reform in the nano-dots. If the magnetic hysteretic is the key factor for the stabilization of vortex at $E = 0$ kV/cm, the out-of-plane single domain would transform into the stripe domain or skyrmions after decreasing the external magnetic field to zero (see Fig. R5b). This feature further confirms that the dominated factor for the stabilization of vortex at $E = 0$ kV/cm is not the magnetic hysteretic effect but the large remanent strain from the ferroelectric substrate. In the case of the non-volatile feature of stripe domain, since such a magnetic state is the ground state in the switching process, it is natural for us to observe that the stripe domain is retained after an E pulse of -10 kV/cm (the stripe domain forms at -10 kV/cm). In the revised manuscript, we have added more discussions on the non-volatile feature (see Pages 8-9, lines 218-230, 234-243, Supplementary Figs. S12-13) and the related references have also been cited (see References 5,9,39,40).

References:

- [1] Caretta, L. et al. Fast current-driven domain walls and small skyrmions in a compensated ferrimagnet. *Nat. Nanotech.* **13**, 1154-1160 (2018).
- [2] Hou, Z. et al. Manipulating the Topology of Nanoscale Skyrmion Bubbles by Spatially Geometric Confinement. *ACS Nano* **13**, 922 (2019).
- [3] Ho, P. et al. Geometrically Tailored Skyrmions at Zero Magnetic Field in Multilayered Nanostructures. *Phys. Rev. Appl.* **11**, 024064 (2019).
- [4] Milde, P. et al. Unwinding of a skyrmion lattice by magnetic monopoles. *Science* **340**, 1076 (2013).
- [5] Peng, L. et al. Real-Space Observation of Nonvolatile Zero-Field Biskyrmion Lattice Generation in MnNiGa Magnet. *Nano Lett.* **17**, 7075 (2017).
- [6] Karube, K. et al. Robust metastable skyrmions and their triangular-square lattice structural transition in a high-temperature chiral magnet. *Nat. Mater.* **15**, 1237 (2016).

Figure R4 a. The remanent strain of $[\text{Pt}/\text{Co}/\text{Ta}]_8$ heterostructure for different electric field (E). The remanent strain is measured after decreasing E from a target value to zero. **b.** Magnetic field-dependent domain evolution process of the $[\text{Pt}/\text{Co}/\text{Ta}]_{12}$ heterostructure. **c.** Magnetic field-dependent domain evolution process of the $[\text{Pt}/\text{Co}/\text{Ta}]_{12}$ continuous thin film.

Figure R5 a. Magnetic field-dependent domain evolution process of the $[\text{Pt}/\text{Co}/\text{Ta}]_8$ heterostructure. These images are taken at $E = 0$ kV/cm after decreasing E from +10 kV/cm. **b.** Magnetic field-dependent domain evolution process with the magnetic field decreasing from 100 mT to 0 mT.

Referee A's Comment #4: (2) Non-volatile. The authors claim that their method is superior over previous results because it is non-volatile: ‘the strain that is associated with the FE polarization is non-volatile, which makes such manipulation of skyrmions non-volatile too

(line 99), and ‘We proposed that such non-volatile switching closely related to both the large remnant strain and the geometrically confined effect’ (line 220). How can you exclude that there is also here charge accumulation at the interface as a driving force? How can you disentangle from magnetic state remanence (see previous question)? (this is also connected to my general question above whether the strain is remanent and what the virgin state is).

Authors’ Reply: We thank the reviewer for his/her valuable comments. In our experiments, the heterostructure is comprised of (001)-oriented PMN-PT, Ta layer with a thickness of 10 nm, and [Pt/Co/Ta]_n multilayer, as shown in Fig. 1a in the main text. The external electrical field is applied on the PMN-PT substrate through the top Ta layer and the bottom Au electrode at the lower surface of PMN-PT. Hence, the charge can only accumulate at the interface between PMN-PT and Ta layer. Meanwhile, the accumulation thickness of the charge is generally less than 1 nm due to a shield effect^[1]. The screening lengths can be calculated by the free-electron model^[1]:

$$\lambda \cong 0.5\left(\frac{n}{a^3}\right)^{-1/6}, \quad (1)$$

where n is carrier density, and $a = 4\pi\hbar^2\varepsilon_0/(m_e e^2)$ is Bohr radius (a is a constant and is equal to 52.9177 pm). For the metals, their carrier concentration is generally in the order of 10^{22}cm^{-3} and the corresponding screening length is calculated to be less than 1 nm^[2,3]. Such a length is much smaller than the thickness of the Ta layer (10 nm) between PMN-PT and magnetic multilayer. Hence, we propose that the charge accumulation can not affect the magnetic multilayer that is deposited on the top of the Ta layer.

On the other hand, we propose that the non-volatile feature of skyrmions originates from the geometrically pinning while the large remnant strain of FE substrate plays a dominate role in the non-volatile feature of vortex state. Detailed discussions on this point are given in our response to the reviewer’s comment #3.

References:

- [1] Ibach, H. & Lüth, H. *Solid State Physics* (Springer, New York, 2009).
- [2] Kim, D. J. et al. Polarization Relaxation Induced by a Depolarization Field in Ultrathin Ferroelectric BaTiO₃ Capacitors. *Phys. Rev. Lett.* **95**, 237602 (2005).
- [3] Pantel, D. & Alexe, M. Electroresistance effects in ferroelectric tunnel barriers. *Phys. Rev. B* **82**, 134105 (2010).

Referee A's Comment #5: (3) Magnetic field. Did the magnetic field from the MFM tip have an influence on the magnetic state? This could be identified by either tip-sample distance dependent measurements or measurements with opposite direction of the external magnetic field.

Authors' Reply: We thank the reviewer for his/her valuable comments. Based on the reviewer's comments, we have investigated the effect of the tip-sample distance on the stabilization of skyrmions. Fig. R6a shows a typical MFM image that is taken at $E = +3$ kV/cm for the [Pt/Co/Ta]₁₂ heterostructure. During the measurement process, no external magnetic field is applied and the distance between the MFM tip and the sample is fixed to be 30 nm (the same distance with that in the main text). One can notice that the skyrmions are stabilized in the nano-dots. Hereafter, we have taken MFM images by varying the tip-sample distances while keeping E still (see Fig. R6b). We find that the variation of tip-sample distance shows little influence on the stabilization of skyrmions though the MFM images become unclear when the tip-sample distance is above 80 nm. This feature suggests that the stabilization of skyrmions is little affected by the magnetic field from MFM tip. On the other hand, we have taken MFM images by using tip that is magnetized with an opposite magnetic field, namely, the direction of the magnetic field used to magnetize the tip is opposite to that of the previous one. As shown in Fig. R6c, the corresponding color map is completely opposite to that in Figs. R6a and 6b. Since the magnetic field of MFM tip has been reversed, the opposite color map suggests that the spin direction of the magnetic states is not reversed by the magnetic field from MFM tip. Meanwhile, we find the morphology of the magnetic states is also slightly affected, which further confirms that the stabilization of skyrmions is little affected by the magnetic field from MFM tip. In the revised manuscript, we have added more discussions on this point (see Page 7, lines 182-184; Supplementary Fig. S6).

Figure R6 a. MFM image taken at $E = +3$ kV/cm under zero magnetic field for the $d \sim 350$ nm nano-dots of the $[\text{Pt}/\text{Co}/\text{Ta}]_{12}$ heterostructure. The distance between MFM tip and sample is fixed to be 30 nm. **b.** MFM image taken at $E = +3$ kV/cm with different tip-sample distance. **c.** MFM images taken at different tip-sample distance by reversing the direction of the magnetic field.

Referee A's Comment #6: (4) Anisotropy. Is the way to calculate K_{eff} as explained in Fig. S10 a typical way to do this? For which kind of systems is this method valid? As I understand there are many different magnetic phases that appear while sweeping the external magnetic field that contribute to the measured total magnetization. One can see that the hysteresis loops do not have the typical S-shape of a ferromagnet but instead reflect transitions between different magnetic phases. Is the evaluation regarding K_{eff} still valid for such a kind of more complex system?

Authors' Reply: We thank the reviewer for raising this valuable question. For magnetic thin films, magnetic anisotropy energy (K) or so called effective magnetic anisotropy energy (K_{eff}), represents the difference of total magnetization work that the external magnetic field does along the perpendicular (W_{\perp}) and parallel $W_{//}$ directions, i.e., $K = W_{\perp} - W_{//}$ ^[1-3]. (1)

To calculate K , the value of W should be established first. On basis of the definition of W , it can be calculated by using the equation ^[1]:

$$W = \int_0^{M_s} \mu_0 H dM, \quad (2)$$

where M represents the magnetic moment under an external magnetic field (H) and the integration interval of M ranges from zero to the saturation magnetization (M_s). On basis of the equation (2), the value of W can be obtained by integrating the area enclosed by the M - H loop (see Fig. R7a) if W completely transform into the magnetic energy. By further combining

the equations (1) with (2), the value of K is hence equal to the area difference enclosed by the M - H loops along perpendicular and parallel directions (see Fig. R7b). As known to us, during the process that the magnetic film is magnetized from virgin state to saturation, the magnetization increases either by domain rotation or by the domain wall motion, or both^[1]. Although different cases result in different shapes of M - H curves, the value of W can be calculated by integrating the area enclosed by the M - H loop if W completely transforms into the magnetic energy that can be stored by the crystal^[1]. For the Pt/Co/Ta multilayered thin film in our experiments, the magnetic field induces a series of transitions between different magnetic phases via the domain wall motion due to a dedicate interplay of different magnetic energies (i.e., ferromagnetic exchange energy, DM energy, magnetic anisotropy energy, demagnetization energy, and Zeeman energy). This feature makes that the area enclosed by the M - H loop is equal to the total magnetization work done by the external magnetic field though the hysteresis loops in our experiments do not have the typical S -shape of a ferromagnet. Hence, the corresponding value of K is equal to the area difference enclosed by the M - H loops along perpendicular and parallel directions.

In some previous literatures on skyrmions^[4-6], K is calculated by using the equation:

$$K = \frac{M_s \mu_0 H_k}{2}, \quad (3)$$

where H_k represents the anisotropy field that is approximately equal to the value of saturation field and M_s represents the saturation magnetization. Indeed, this equation can be considered as a simplified form of equation (2). As shown in Fig. R7c, if the M - H curves along perpendicular and parallel directions are straight lines, the shape of the area enclosed by the M - H loops is triangular and hence the value of K can be calculated by using the triangle area formula: $\frac{M_s \mu_0 H_k}{2}$.

However, if the magnetic field induces a structure transition, the area enclosed by the M - H loop is not equal to the total magnetization work done by the external magnetic field because some of the magnetization work transform into the thermal energy or other forms of energy that cannot be reflected by the area enclosed by the M - H loop. In that case, the method in our work is not valid for calculating K .

References:

- [1] Cullity, B. D. & Graham, C. D. *Introduction to magnetic materials*. (John Wiley & Sons, Inc, Hoboken, 2009).
- [2] Heide, M., Bihlmayer, G. & Blügel, S. Dzyaloshinskii-Moriya interaction accounting for the orientation of magnetic domains in ultrathin films: Fe/W (110). *Physical Review B (R)* **78**, 140403 (2008).
- [3] Baberschke, K. The correlation of local structure and magnetism in thin ferromagnetic films. *Surface Review and Letters* **67**, 735 (1999).
- [4] Woo, S. et al. Observation of room-temperature magnetic skyrmions and their current-driven dynamics in ultrathin metallic ferromagnets. *Nat. Mater.* **15**, 501–506 (2016).
- [5] Jiang, W. et al. Direct observation of the skyrmion Hall effect. *Nat. Phys.* **13**, 162–169 (2017).
- [6] Boulle, O. et al. Room-temperature chiral magnetic skyrmions in ultrathin magnetic nanostructures. *Nat. Nanotech.* **11**, 449–454 (2016).

Figure R7 a. Schematic of a typical magnetization cure of a ferromagnet. The area of the shaded part represents the total magnetization work that the external magnetic field does. M_s represents the saturation magnetizations. **b.** Schematic of the typical magnetization cures along the perpendicular direction (hard-magnetization axis) and parallel direction (easy-magnetization axis) of a ferromagnet. The area of the shaded part represents the difference of the total magnetization work done along the perpendicular and parallel directions. **c.** Schematic of two special magnetization cures, where the shape of the area enclosed by the curves (shaded part) is triangular. H_k represents the corresponding magnetic field for the saturation magnetization.

Referee A's Comment #7: The authors state that ‘an out-of-plane single domain started to appear in the nano-dot’ (line 150). Why would the single domain be out-of-plane when there is an in-plane anisotropy?

Authors' Reply: We thank the reviewer for the valuable comments. Our results demonstrate that, as the diameter of the nano-dots decreases from 1 μm to 150 nm, the domain structure gradually transforms from a stripe domain into an out-of-plane single domain (see Fig. 1 in the main text). Notably, our experimental observations are not extraordinary and similar

results have been widely reported in previous reports^[1-3]. Such a diameter-dependent domain evolution process can be attributed to the transition of the magnetic anisotropy from an in-plane type into an out-of-plane one. As known to us, the magnetic anisotropy of the thin film comprise of magneto-crystalline anisotropy (K_m), interface anisotropy (K_i), and shape anisotropy (K_s), namely, $K = K_m + K_i + K_s$ ^[4,5]. Since K_m and K_i are mainly determined by the intrinsic spin-orbit coupling of the magnetic film, we assume that the two physical parameters do not vary with the decrease of the diameter of the nano-dots. However, K_s is significantly affected by the shape variation of the nano-dots. For an ellipsoid (see Fig. R8a), K_s is given by^[4]:

$$K_s = \frac{1}{4}\mu_0 M_s^2 (1 - 3N), \quad (1)$$

where N is the demagnetizing factor along the c -axis and M_s is the saturation magnetization. When a -axis is much larger than that of c -axis ($a \geq c$), the value of N along c -axis is equal to 1. Thus, the value of K_s is calculated to be $-\frac{1}{2}\mu_0 M_s^2$. In this case, K_s shows a negative value, meaning that K_s prefers spin to lie in the in-plane direction of the ellipsoid. The magnetic anisotropy K is hence equal to $K_m + K_i - \frac{1}{2}\mu_0 M_s^2$. With the decrease of the length of a -axis, the value of N gradually decreases and the value of K_s increases correspondingly. When c -axis is much larger than that of a -axis ($a \leq c$), N is equal to 0. The value of K_s is equal to $\frac{1}{4}\mu_0 M_s^2$. In this case, K_s shows a positive value and prefers the spin to orientate along the out-of-plane direction of the ellipsoid. The magnetic anisotropy K is hence equal to $K_m + K_i + \frac{1}{4}\mu_0 M_s^2$. Based on the results above, we can find that the value of K can be significantly affected by the shape variation of the nano-dots. As for our experiments, the cylindrical nano-dot can be approximated to the ellipsoid. For the nano-dot with a diameter of 1 μm , N is proposed to be 1 because the value of diameter is much larger than that of thickness (60 nm). Hence, the value of K is equal to $K_m + K_i - \frac{1}{2}\mu_0 M_s^2$. Our magnetic measurements have demonstrated K of such a nano-dot shows a negative value (see Fig. R8b), meaning that the nano-dot possesses an in-plane anisotropy. With the decrease of diameter, the value of N decreases, which makes both the corresponding absolute values of K_s and K increases towards the positive values,

namely, the decrease of diameter gradually induces the magnetic anisotropy to transform from an in-plane type to an out-of-plane one. When the diameter decreases to 150 nm, the magnetic anisotropy of the nano-dots completely transforms into an out-of-plane type (see Fig. R8c) and hence the out-of-plane single domain is observed.

References:

[1] Ho, P. et al. Geometrically Tailored Skyrmions at Zero Magnetic Field in Multilayered Nanostructures. *Phys. Rev. Appl.* **11**, 024064 (2019).
 [2] Cowburn, R. P., Koltsov, D. K., Adeyeye, A. O., Welland, M. E., & Tricker, D. M. Single-domain circular nanomagnets. *Phys. Rev. Lett.* **83**, 1042 (1999).
 [3] Guslienko, K. Y. Magnetic vortex state stability, reversal and dynamics in restricted geometries. *Journal of nanoscience and nanotechnology* **8**, 2745-2760 (2008).
 [4] Coey, J. M. D. *Magnetism and Magnetic Materials* (Cambridge Univ. Press, 2009).
 [5] Hellman, F. *Encyclopedia of Materials: Science and Technology* (Elsevier, Amsterdam, 2001).

Figure R8 a. Schematic of an ellipsoid. Out-of-plane and in-plane hysteresis loops of the $[\text{Pt}/\text{Co}/\text{Ta}]_{12}$ b thin film and c d -150 nm nanodot. The magnetic anisotropy of thin film shows an in-plane feature while the magnetic anisotropy of d -150 nm nanodot shows a perpendicular feature.

Referee A's Comment #8: The color scale in the simulated images ranges from red over white to blue. In the majority of the presented images the color scale suggests, that the magnetic state is dominantly out-of-plane, this does not seem likely for a system with in-plane anisotropy. How is the color scale set? Is it identical for all images?

Authors' Reply: We thank the reviewer for the valuable comments. In our manuscript, we have simulated the magnetic domain evolution process with the variation of K_{eff} and D to qualitatively explain the physical origin of the experimentally observed strain-induced switching of different magnetic domain states. In the simulations, the out-of-plane component of magnetization is represented by the red (+z)-white (0)-blue (-z) color scale. As known to us,

the spin distribution of a skyrmion-hosting magnetic system is determined by a dedicated interplay of ferromagnetic exchange interaction, magnetic anisotropy, DMI, and external magnetic field. Among these terms, the variation of magnetic anisotropy, DMI, and external magnetic field may significantly affect the out-of-plane component of magnetic domain states. For our sample, although the magnetic anisotropy shows an in-plane feature, the corresponding easy magnetization axis may not completely lie into the in-plane direction. For a thin film, the so called “in-plane anisotropy” is different from the term “in-plane anisotropy” used to describe the magnetic anisotropy of bulk crystal. For a bulk crystal, the in-plane anisotropy means the easy-axis completely lies into the in-plane direction of the sample. However, the case is different for a thin film. Different from the 3D structure of a bulk crystal, the thin film only possesses two characteristic directions, i.e., in-plane and out-of-plane directions. If the in-plane direction of the thin film is more easily magnetized compared to the out-of-plane direction, the magnetic anisotropy is termed as “in-plane anisotropy”. In this case, the easy-axis of the thin film is no need to completely lie into the in-plane direction and may be tilted a certain angle towards the out-of-plane direction. As mentioned above (see response to the referee A’s Comment #7), the magnetic anisotropy of the thin film comprises of magneto-crystalline anisotropy (K_m), interface anisotropy (K_i), and shape anisotropy (K_s), namely, $K = K_m + K_i + K_s$. In the simulations, the value of K is set to be $-0.9 \times 10^5 \text{ J/m}^3$ which is equal to experimentally measured value of d -350 nano-dot, and the value of K_s is set to be $-3.1 \times 10^5 \text{ J/m}^3$ ($K_s = -\frac{1}{2} \mu_0 M_s^2$) by assuming that the demagnetizing factor along the c -axis is equal to 1. Thus, we can calculate the value of $K_m + K_i$ to be $2.2 \times 10^5 \text{ J/m}^3$. Such a positive value of $K_m + K_i$ suggests that the magnetic anisotropy of the sample in our experiments possesses an out-of-plane component, namely, the easy-axis does not completely lie into the in-plane direction but is tilted a certain angle towards the out-of-plane direction. The canted easy-magnetized axis may increase the out-of-plane components of magnetic states significantly, which makes the magnetic domain states in our simulations possess more out-of-plane spin components than that with a completely in-plane easy-axis. Moreover, our sample possesses a large DMI which may also increase the out-of-plane spin components of the magnetic states. Besides, an out-of-plane magnetic field of 20 mT is applied in our

simulations. Such an external magnetic field can force the spin to originate along the out-of-plane direction, which may make magnetic state seem to be dominantly out-of-plane. For example, as shown in Fig. R5a in response to referee A's comment #3, if we apply an external magnetic field of 50 mT along the out-of-plane direction of the nano-dot, the in-plane vortex can transform into an out-of-plane single domain. Thus, an out-of-plane of magnetic field is quite possible to make magnetic state seem to be dominantly out-of-plane, especially in our sample with a weak in-plane anisotropy and strong DMI.

Referee A's Comment #9: (5) DMI. The authors state that 'Based on the values of σ_{DW} , A , and K_{eff} , we derived the absolute value of D ' (line 461). The σ_{DW} (domain wall energy) was derived from the domain widths as explained in Fig. S11. Here the authors use simulations with some material parameter input values to simulate MFM images which are then compared to the MFM experimental data to determine the wall width. This method is not a reasonable way to obtain these values. Thus also the derived σ_{DW} and thus also the derived DMI are not reliable. The period of the stripe domain is a measure for the DMI: for a given A and K the DMI sets the length scale; the domain wall width is independent from the DMI, only the domain wall energy depends on the DMI (then still the problem remains how to determine K and whether A can be treated as strain-independent).

Authors' Reply: We thank the reviewer for the valuable comments. These comments are greatly helpful to improve the quality of our manuscript. In this work, we derive the absolute value of D on basis of the values of σ_{DW} , K_{eff} , and A . The value of σ_{DW} can be quantified by measuring the low-magnetic-field domain period (w) on the basis of the classical domain spacing model:

$$\frac{\sigma_{DW}}{\mu_0 M_S^2 t} = \frac{w^2}{t^2} \sum_{odd\ n=1}^{\infty} \left(\frac{1}{(\pi n)^3} \right) [1 - (1 - 2\pi n t/w) \exp(-2\pi n t/w)], \quad (1)$$

where t is the thickness of the film, M_S is the saturation magnetization, and w is the low-field domain period ($w = w_{\uparrow} + w_{\downarrow}$, w_{\uparrow} and w_{\downarrow} were the up and down domain widths, respectively) that can be directly measured from the MFM image. As known to us, a domain structure comprises of the domain wall and domain, as shown in Fig. R9a. In our initial manuscript, we proposed that the value of w in the equation was equal to that of the domain width excluding

the domain wall width (see Fig. R9a). To separate the domain wall from the domain structure, we first simulated the MFM image of a Néel-type stripe domain and subsequently investigated the corresponding contrast variations of the domain wall and domain in a simulated MFM image. Having roughly known the contrast variations of domain wall and domain in a MFM image, we hereafter measured their values on the experimental MFM images. On basis of the reviewer's comments, we have re-checked the equation and find that the value of w should be equal to the domain period that includes both the domain wall and domain. Thus, we have recounted the values of w by directly measuring the period of the stripe domain on the experimental MFM images. Fig. R9b shows the typical contrast variation of a domain structure. We propose that the width between two neighbor contrast peaks is equal to the period of a domain structure and their corresponding values at different electrical field (E) are summarized in the Supplementary Table S2. Notably, the position of contrast peak is established by using a Gauss fitting and the error bar of the period is added by measuring the periods at different regions of a MFM image. On basis of recounted values of w , the corresponding values of σ_{DW} at different electrical fields are re-calculated and also summarized in the Supplementary Table S2.

On basis of the definition of magnetic anisotropy energy (K), K in a magnetic film represents the difference of total magnetization work done along the perpendicular (W_{\perp}) and parallel $W_{//}$ directions, i.e., $K = W_{\perp} - W_{//}$. The value of W can be obtained by integrating the area enclosed by the M - H loop (see Fig. R7a) if W completely transform into the magnetic energy. For the Pt/Co/Ta multilayered thin film in our experiments, the magnetic field induces a series of transitions between different magnetic phases via the domain wall motion. As known to us, such transitions completely originate from a dedicate interplay of different magnetic energies (i.e., ferromagnetic exchange energy, DM energy, magnetic anisotropy energy, demagnetization energy, and Zeeman energy). This feature makes that the area enclosed by the M - H loop is equal to the total magnetization work done by the external magnetic field though the hysteresis loops in our experiments do not have the typical S -shape of a ferromagnet. Hence, the corresponding value of K is equal to the area difference enclosed by the M - H loops along perpendicular and parallel directions. More discussions on this point

have been presented in the response to the reviewer's comment #6. The E -dependent K_{eff} can be further obtained by measuring the E -dependent magnetization curves of the heterostructure along both the in-plane and out-of-plane directions and we have summarized E -dependent K_{eff} of the [Pt/Co/Ta]₁₂ heterostructure in Supplementary Table S2. The error bar is added by measuring K_{eff} of two different [Pt/Co/Ta]₁₂ heterostructures.

In our work, we have measured the temperature-dependent saturation magnetization $M_s(T)$ to establish the value of A (see Fig. R10). The $M_s(T)$ curve can be fitted with the Bloch law^[1,2]:

$$M_s(T) = M_0(1 - b/T^{3/2}), \quad (2)$$

where M_0 is the saturation magnetization at zero Kelvin and b is a constant. The constant b can be further given by:

$$A = \frac{nk_B S^2}{a} \left(\frac{C}{b}\right)^{2/3}, \quad (3)$$

where $n = 4$ is the coordination number for a fcc lattice^[1,2] (the Co that deposited at room-temperature using sputter generally crystallizes into the fcc structure^[2,3]), k_B is the Boltzmann constant that takes a value of 1.380649×10^{-23} J/K, S is the spin per atom that takes the value of 1^[2,4], C is a constant that takes the value 0.0294 for an fcc lattice^[1,2], and a is the lattice constant of fcc Co that takes a value of 0.3536 nm^[2,5]. Thus, if the value of b is established, we can directly calculate the value of A by using equation (3). To determine the value of b , we have fitted the temperature-dependent saturation magnetization of the [Pt/Co/Ta]₁₂ heterostructure at zero electrical field with the Bloch law, as shown in Fig. R10. We find that the $M_s(T)$ curve agrees well with the Bloch law over the temperature range of 60 K-330 K. However, when the temperature decreases below 60 K, the curve deviates from the Bloch law. Such a deviation may be attributed to the pronounced role of the Ruderman-Kittel-Kasuya-Yosida (RKKY) interaction at the low-temperature range (below 60 K)^[6,7]. The corresponding value of A can be calculated to be 16.9 ± 0.2 pJ/m. The error bar is added by fitting the $M_s(T)$ curve within different temperature range and such a value of A falls within the reported range of A for the Co-based thin film^[4,7,8].

To investigate the effect of strain on the variation of A , we have measured the temperature-dependent saturation magnetization of the [Pt/Co/Ta]₁₂ heterostructure at +10

kV/cm where the PMN-PT substrate exhibits the maximum strain. As shown in Fig. R10, we can find that the $M_s(T)$ curve keeps nearly unchanged at the whole temperature range of 10-330 K compared with that measured at a zero electrical field. Thus, the value of A should keep nearly unchanged under such a strain. That is to say, the value of A is insensitive to the strain in our experiments. On the other hand, previous experimental reports demonstrated that when a strain of 0.3% was applied on a FeGe thin plate, the value of T_c just varied by 3.3 K^[9]. Such a slight variation of T_c only corresponds to a few percentage variation of A compared with that at the ambient pressure ($T_c = 280$ K)^[9]. For the [Pt/Co/Ta]₁₂ heterostructure in our experiments, the maximum tensile strain is 0.026% and the maximum compressive strain is 0.07%. These values are nearly one order of magnitude lower than that applied on the FeGe. Thus, we believe that the value of A is slightly affected by such a strain and it is reasonable for us to treat A as a strain-independent parameter in our experiments. In the revised manuscript, we have added more discussions on this point (see Page 10, lines 267-268; Page 13, lines 365-382; Supplementary Fig. S14).

References:

- [1] Chikazumi, S. *Physics of Ferromagnetism* (Oxford University Press, Oxford, 1997).
- [2] Shahbazi, K. et al. Domain-wall motion and interfacial Dzyaloshinskii-Moriya interactions in Pt/Co/Ir(t_{ir})/Ta multilayers. *Phys. Rev. B* **99**, 094409 (2019).
- [3] Wells, A. W., Shepley, P. M., Marrows, C. H. & Moore, T. A. Effect of interfacial intermixing on the Dzyaloshinskii-Moriya interaction in Pt/Co/Pt. *Phys. Rev. B* **95**, 054428 (2017).
- [4] Balashov, T. et al. Magnetic anisotropy and magnetization dynamics of individual atoms and clusters of Fe and Co on Pt (111). *Phys. Rev. Lett.* **102**, 257203 (2009).
- [5] Liu, X. et al. Exchange stiffness, magnetization, and spin waves in cubic and hexagonal phases of cobalt. *Phys. Rev. B* **53**, 12166 (1996).
- [6] Tahiri, N., Ez-Zahraouy, H. & Benyoussef, A. Multilayer transition in a spin 3/2 Blume–Capel model with RKKY interaction. *Physica A* **388**, 3426-3432 (2009).
- [7] Knepper, J. W. & Yang, F. Y. Oscillatory interlayer coupling in Co/Pt multilayers with perpendicular anisotropy. *Phys. Rev. B* **71**, 224403 (2005).
- [8] Talapatra, A. & Mohanty, J. Laser induced local modification of magnetic domain in Co/Pt multilayer. *J. Magn. Magn. Mater.* **418**, 224-230 (2016).
- [9] Shibata, K. et al. Large anisotropic deformation of skyrmions in strained crystal. *Nature Nanotech.* **10**, 589 (2015).

Figure R9 a. An amplification MFM image. In this image, the regions that represent the domain wall, up domain and down domain are clearly marked. **b.** The contrast variation extracted from the MFM image. The distance (L) between two neighboring contrast peaks is proposed to be a domain period (w). The experimental data is fitted by Gauss function (red line) to establish the position of the peak.

Figure R10 Temperature-dependent saturation magnetization $M_s(T)$ of the $[\text{Pt}/\text{Co}/\text{Ta}]_{12}$ heterostructure at $E = 0$ kV/cm (orange dots) and $+10$ kV/cm (blue dots). The experimental dots are fitted by using the Bloch law (red line).

Referee A's Comment #10: Vortex This MFM-images that are interpreted as vortex states do not look like what I expect for a vortex measured with MFM, see Shinjo et al., Science 289, 930 (2000). Why do you not see the out-of-plane region in the vortex in MFM? Which material parameters do the spin textures in Fig. S6 correspond to, in particular how large is the vortex core and is that reasonable?

Authors' Reply: We thank the reviewer for the valuable comments. These comments are great helpful to improve the quality of our manuscript. In the d -350 nm nano-dots of the $[\text{Pt}/\text{Co}/\text{Ta}]_8$ heterostructure, we found that the contrast of the stripe or skyrmions decreased

with the increase of compressive strain. This feature suggests that the compressive strain forces the spins to gradually align with the in-plane direction to form an in-plane domain. At $E = +10$ kV/cm that corresponds to the maximum compressive strain, the contrast of the magnetic domains in the nano-dots becomes quite similar to the substrate, which indicates that the magnetic states have transformed into the in-plane domain. To confirm this point, we have further investigated the strain-induced domain evolution process in the $[\text{Pt}/\text{Co}/\text{Ta}]_8$ continuous thin film. As shown in Fig. R11, the stripe domain gradually transforms into the in-plane domain with the increase of electrical field, which is similar to the observations in the nano-dots. This feature confirms that the strain-induced formation of in-plane domain observed in the nano-dots is reasonable.

Having established that the compressive strain can induce the formation of in-plane magnetic domain in the $d=350$ nm nano-dots, we hereafter investigate the spin texture of the in-plane domain. First, we have simulated the dependence of the domain evolution process on the variation of magnetic anisotropy by using a micromagnetic method, as shown in Fig. 4d in the main text. With the increase of in-plane anisotropy, the stripe domain or skyrmion gradually transforms into the vortex state. The formation of vortex state in the nano-dots can be attributed to the geometrically confined effect that makes the vortex state possesses the minimum energy in contrast to the other types of in-plane magnetic states. On basis of these simulations above, we propose that the in-plane magnetic domain in the nano-dots may be the vortex state.

For a regular vortex state, the corresponding MFM image generally shows a core in the central region of the nano-dot^[1-3]. In our experiments, we have observed two types of MFM images, as shown in Fig.R12. For the type-I MFM image (see Fig. R12), we can observe a core in the central region of the nano-dot though the core size in our MFM image is larger than that reported by Shinjo *et al*^[1]. Based on our experimental magnetic parameters (see Table R2), we simulated the MFM image of vortex, as shown in Fig. R12. In such a simulation, the tip-sample distance is fixed to be 30 nm that is the same with that in our experiments. We can find that the MFM image is similar to our experimental results, indicating that the corresponding domain state of type-I MFM image is the vortex state. We

also find that the core size decreases correspondingly with the increase of in-plane anisotropy (decrease the value of K_{eff}), as shown in Fig. R13. For the Ni-Fe film^[1], the easy axis completely lies into the in-plane direction. However, in the case of our nano-dot, the easy axis is tilted a certain angle towards the out-of-plane direction though the corresponding magnetic anisotropy is established to be in-plane (see response to referee A's comment #8). Thus, we propose that the smaller core size in the work reported by Shinjo *et al.*^[1] can be attributed to the stronger in-plane anisotropy. Moreover, we also simulated the MFM image by varying the tip-sample distance (see Fig. R12). We find that the contrast of core decreases correspondingly with the increase of tip-sample distance. In some of the type-I MFM images, e.g., the MFM image shown in the Supplementary Fig. S8 and Fig. R12, the contrast of core is vague. The vague contrast may be attributed to a higher sample-tip distance because the thickness of the nano-dots may be non-uniform.

For the type-II MFM image (see Fig. R12), the core is completely unable to be observed and a white-black contrast variation appears at the edge region. On basis of previous reports^[4-7], such a MFM image is quite similar to that of the distorted vortex state, where the core of the spin texture deviates from the central region. In our manuscript, we have also simulated the MFM images of different types of distorted vortex. As shown in Fig. R12 and Fig. R14, if the core is near the edge of the nano-dot, a clear white-black contrast variation appears at the edge region and the core may be overlapped by such a contrast. By comparing the simulated image with the experimental one, we find that the type-II image agrees well with that shown in Fig. R12. Thus, we can conclude that the type-II MFM image corresponds to the distorted vortex state where the core is close to the edge of the nano-dot. The formation of distorted vortex states in our experiments is reasonable because the strain distribution on the nano-dot is non-uniform, as shown in Fig. R1. The existence of the strain gradient may force the core of the vortex state to orientate along a certain direction^[8]. In the revised manuscript, we have added more discussions on this point (see Page 7, lines 200-202; Supplementary Fig. S8) and the related references are also cited (see references 37,38).

References:

[1]Shinjo, T., Okuno, T., Hassdorf, R., Shigeto, K. & Ono, T. Magnetic vortex core observation in circular dots of permalloy. *Science* **289**, 930-932 (2000).

- [2] Shima, H. et al. Pinning of magnetic vortices in microfabricated permalloy dot arrays. *J. Appl. Phys.* **92**, 1473 (2002).
- [3] Zhu, X. et al. Magnetization reversal and configurational anisotropy of dense permalloy dot arrays. *Appl. Phys. Lett.* **80**, 4789 (2002).
- [4] Buda, L. et al. Vortex States Stability in Circular Co(0001) Dots. *IEEE Trans. Magn.* **37**, 2061 (2001).
- [5] Dai, Z. M. et al. Magnetization reversal of vortex states driven by out-of-plane field in the nanocomposite Co/Pd/Ru/Py disks. *Appl. Phys. Lett.* **111**, 022404 (2017).
- [6] Natali, M. et al. Correlated Magnetic Vortex Chains in Mesoscopic Cobalt Dot Arrays. *Phys. Rev. Lett.* **88**, 157203 (2002).
- [7] Pokhil, T. Spin vortex states and hysteretic properties of submicron size NiFe elements. *J. Appl. Phys.* **87**, 6319 (2000).
- [8] Ostler, T. A. et al. Strain Induced Vortex Core Switching in Planar Magnetostrictive Nanostructures. *Phys. Rev. Lett.* **115**, 067202 (2015).

Figure R11 Electrical-field dependent MFM images of the $[\text{Pt}/\text{Co}/\text{Ta}]_8$ heterostructure. The stripe domain gradually transform into the in-plane domain with the increase of electrical field.

Figure R12 Experimental MFM image (the top row), simulated MFM image (the middle row), and corresponding spin texture (the bottom row) for vortex and distorted vortex. The term “Height” represents the distance between sample and tip.

K_{eff} ($\times 10^5 \text{ J/m}^3$)	A ($\times 10^{-11} \text{ J/m}$)	D (mJ/m^2)	d (nm)	h (nm)
-1.5	1.70	1.0	380	30

Table R2 The magnetic parameters used for simulated the vortex state. h represents the distance between the sample and tip.

Figure R13 The variation of spin texture, core size, and simulated MFM image of vortex with the decrease of K_{eff} .

Figure R14 Upper panel shows the different types of vortices with their core deviated from the central region (upper panel). Lower panel shows the simulated MFM images based on the vortex above.

Referee A's Comment #11: The authors mention an energy barrier between the skyrmion and the vortex (line 209), but why would there be an energy barrier when that is a smooth continuous transition!?

Authors' Reply: We thank the reviewer for the valuable comments. The suggestions and comments are greatly helpful for us to improve the quality of our manuscript. Our experiments have demonstrated that the skyrmion continuously transforms into the vortex state as the increase of compressive strain. However, the skyrmion possesses a topological number of 1 while the vortex possesses a topological number of 0.5. The difference of the topological structure between skyrmion and vortex may lead to the existence of an energy barrier. However, since the experimentally observed skyrmion-vortex transition is nearly

continuous, we believe that such an energy barrier is quite low. To avoid possible misleading of readers, we use the term “energy difference” to replace “energy barrier” in the revised version (see Pages 7-8, lines 206-209).

Referee A’s Comment #12: How can geometry favor a vortex state over a skyrmion state? Either there is in-plane or out-of-plane anisotropy, that should be the decisive parameter to select one of the two.

Authors’ Reply: We thank the reviewer for the valuable comments. At initial state ($E = 0$ kV/cm and $\varepsilon = 0$), the sample shows an in-plane magnetic anisotropy (IMA). We should note here that such an IMA skyrmion-hosting film has also been reported in previous literatures^[1-4] and meanwhile the skyrmion density in the IMA film is generally larger than that in the film with perpendicular magnetic anisotropy (PMA) due to a dedicate interplay of different magnetic energies^[1-4].

As E increases from 0 to +3 kV/cm, the tensile strain induces the stripe domain to gradually convert into the skyrmion. As we further increase E above +3 kV/cm, the strain gradually transforms into the compressive type and induces the skyrmion to transform into the vortex. Our experimental results demonstrate that, with the increase of compressive strain, the value of DMI constant (D) increases while the value of K decreases from $-0.9 \times 10^5 \text{ J/m}^3$ (the value of K is negative for an IMA) at $E = 0$ kV/cm to $-1.1 \times 10^5 \text{ J/m}^3$ at $E = +10$ kV/cm. Based on the simulations (see Fig. 4c in the main text), we find that the increase of D cannot induce the skyrmion to transform into the vortex, namely, D is not the decisive magnetic parameter for the skyrmion-vortex transition. However, if the value of K decreases, the skyrmion gradually converts into the vortex (see Fig. 4d in the main text). This feature suggests that the increase of IMA is the key factor for the skyrmion-vortex transition. Moreover, we find that the magnetic anisotropy remains an in-plane type during the skyrmion-vortex transition, which further suggests that a variation of magnetic anisotropy from PMA to IMA is not the necessary condition for the skyrmion-vortex transition in our experiments and only an increase of IMA (magnetic anisotropy keeps an IMA feature) may also result in such a transition.

References:

- [1] He, M. et al. Evolution of topological skyrmions across the spin reorientation transition in Pt/Co/Ta multilayers. *Phys. Rev. B* **97**, 174419 (2018).
- [2] Zhang, S. et al. Determination of chirality and density control of Néel-type skyrmions with in-plane magnetic field. *Communications Physics* **1**, 36 (2018).
- [3] Wang, L. et al. Construction of a Room-Temperature Pt/Co/Ta Multilayer Film with Ultrahigh-Density Skyrmions for Memory Application. *ACS Appl. Mater. Interfaces* **11**, 12098 (2019).
- [4]Güngörduü, U. et al. Stability of skyrmion lattices and symmetries of quasi-two dimensional chiral magnets. *Phys. Rev. B* **93**, 064428 (2016).

Referee A's Comment #13: (7) Color scale The color bar used for the experimental data is not clear, also to compare images of the same series (i.e. identical magnetic tip and lift mode height) the color scale should be set identical for all measurements to allow better comparison (or at least stated how the color scale was set).

Authors' Reply: We thank the reviewer for the valuable comments. In the revised manuscript, the color bar of the MFM images has been added for better comparison. We should note here that, although the magnetic tip and lift mode height are identical in our experiments, the thickness of the nano-dot is different in the [Pt/Co/Ta]₈ and [Pt/Co/Ta]₁₂ heterostructures. Since the thickness variation may also affect the MFM signal significantly^[1,2], we set different color bar for the MFM images of the nano-dots with different thickness, i.e., the color scale is set range from -2.3 Hz to 2.3 Hz for the [Pt/Co/Ta]₁₂ nano-dot, and the color scale is set range from -1.4 Hz to 1.4 Hz for [Pt/Co/Ta]₈ nano-dot (see Fig. 1c, Fig. 2, and Fig. 3a).

References:

- [1] Van Schendel, P., Hug, H., Stiefel, B., Martin, S. & Güntherodt, H.-J. A method for the calibration of magnetic force microscopy tips. *J. Appl. Phys.* **88**, 435–445 (2000).
- [2] Joshi, N. R., Özer, S. & Hug, H. J. Engineering the ferromagnetic domain size for optimized imaging of the pinned uncompensated spins in exchange-biased samples by magnetic force microscopy. *App. Phys. Lett.* **98**, 082502 (2011).

Response to the Report of Referee B

Referee B's General Comment: Authors investigated electric-field manipulation of individual skyrmions in a nanostructured ferromagnetic/ferroelectrical heterostructure at room temperature via an inverse magneto-mechanical effect. In the present research work the authors showed that such manipulation of skyrmions is non-volatile and exhibits a multi-state feature. Micromagnetic simulation results indicate that the electric-field manipulation of skyrmions originates predominantly from the strain-mediated modification of effective magnetic anisotropy and Dzyaloshinskii-Moriya interaction.

The paper is well organized and the presentation of the results is clear. The conclusions are soundly supported by data presented in the manuscript and in the supplementary data. The paper is very interesting, well written and provides concise explanations, justifications and physical insights towards new directions regarding the construction of low energy dissipation, non-volatile, and multi-state skyrmion-based spintronic devices.

I recommend this paper for publication in "Nature Communications" subject to a number of minor changes and corrections that could be addressed.

Authors' Reply: We sincerely thank the reviewer for careful reading of our manuscript and for recommending our manuscript to be published in Nature Communications. The valuable suggestions and comments are greatly helpful to improve our manuscript. Below we answer the reviewer's questions and comments in a point-by-point basis. We hope the reviewer will be satisfied with the revised manuscript as well as our responses.

Referee B's Comment #1: The authors should try to shorten the lengthy Introduction section (compared to the actual manuscript's presentation discussion; approximately 2 pages from a total of 11 pages). In particular, the authors should trim down lines 103-119 of page 4. The same discussion naturally belongs to the summary which is nicely given by the authors in pages 11-12 (lines 320-329).

Authors' Reply: We thank the reviewer for the valuable comments. In the revised manuscript, the introduction section has been shortened (see Page 4, lines 109-113).

Referee B's Comment #2: Page 4 line 100: stimulated.

Authors' Reply: We thank the reviewer for careful reading of our manuscript. In the revised version of our manuscript, this point has been revised (see Page 4, line 99).

Referee B's Comment #3: The present study and the experimental procedure took place at room temperature. The micromagnetics simulations presented in Methods Section express the energy density having contributions from exchange, anisotropy, demagnetization and Dzyaloshinski-Moriya. No thermal field is introduced or implicitly stated. Presumably, the micromagnetic simulations took place at 0 K. The authors should comment on this in a more detailed manner.

Authors' Reply: We thank the reviewer for pointing out this problem. Yes, our micromagnetic simulations are performed at zero temperature. We have updated the simulation details given in the method section according to the reviewer's suggestion (see Pages 12-13, lines 356-357).

Referee B's Comment #4: Page 10 line 279. The figure 4b should be corrected. The epsilon-dependent D refers to figure 4a.

Authors' Reply: We thank the reviewer for careful reading of our manuscript. In the revised version of our manuscript, the order of Figure 4a and 4b are exchanged (see Page 20, Figure 4a and 4b).

Response to the Report of Referee C

Referee C's General Comment: The article describes how the magnetic texture in nano discs is modified under strain. The strain is modulated using an electric field on a piezoelectric (also ferroelectric) PMN-PT material. The magnetization texture can switch from in-plane magnetization to out-of-plane hosting worm domain or skyrmions, depending upon electric, magnetic field, and disc diameter. These experimental results are impressive.

The English is globally good, but all verbs are in past tense, which read very strange sometimes: e.g. "The details of the simulations were provided in the Methods section". They are still in this section... I am shocked by the use of different types of units, sometimes even in the same sentence (ll.447-448) or figure (4): erg, J, V, Oe, mT, ... Authors should stick to modern SI units! Because of this poor consistency, the reader does not know a priori if the formula in the text are expressed in the cgs/emu or SI unit system (or even worse, a mix of them!?). Punctuation after equations should be revised. Figures are OK, except for the units or other details.

I think that this article deserves a bit more work before being published. The results are already fine and interesting, but their interpretation and description in the article could be improved substantially by a few more experiments. I suggest a list of possible improvements below.

Authors' Reply: We sincerely thank the reviewer for careful reading of our manuscript and for pointing out that our experimental results are impressive. Following the referee's comments and suggestions, we have carried out additional experiments and analysis to improve the interpretation and description of the manuscript. Moreover, the tense and units of the manuscript have also been revised thoroughly. Below, we address the referee's comments and questions in a point-by-point basis. We hope the referee is satisfied with the revised manuscript and our response.

Referee C's Comment #1: lines 115-116 + 141-145: "except a low [field] of less than 100 Oe from the MFM tip": In fact this field is applied only when the tip flies over the surface (for topography or MFM). I do not believe that the skyrmion is written by the tip while measuring the disc. Usually, the opposite phenomenon happens: the skyrmion is deleted by the tip.

Authors could check if the skyrmion changes shape when reversing the magnetization of the MFM tip! This would avoid useless (?) lengthy comments about the tip stray field.

Authors' Reply: We thank the reviewer for the valuable comments. The suggestions and comments are greatly helpful for us to improve the quality of our manuscript. To exclude the influence of the tip stray field on the domain evolution process, we have first investigated the effect of the tip-sample distance on the stabilization of skyrmions. Fig. R1a shows a typical MFM image taken at $E = +3$ kV/cm for the [Pt/Co/Ta]₁₂ heterostructure. During the measurement process, no external magnetic field is applied and the distance between the MFM tip and the sample is fixed to be 30 nm (the same distance with that in the main text). One can notice that the skyrmions are stabilized in the nano-dots. Hereafter, we have taken MFM images by varying the tip-sample distances while keeping E still (see Fig. R1b). We find that the variation of tip-sample distance shows little influence on the stabilization of skyrmions though the MFM images become unclear when the tip-sample distance is above 80 nm. This feature suggests that the stabilization of skyrmions is little affected by the magnetic field from MFM tip. On basis of the reviewer's comments, we have further taken MFM images by using tip that is magnetized with an opposite magnetic field, namely, the direction of the magnetic field used to magnetize the tip is opposite to that of the previous one. As shown in Fig. R1c, the corresponding color map is completely opposite with that in Figs. R1a and 1b. This feature suggests that the spin direction of the magnetic states is not reversed by the magnetic field from MFM tip. Meanwhile, we find the morphology of the magnetic states are also little affected. This feature further confirms that the stabilization of skyrmions is little affected by the magnetic field from MFM tip. In the revised manuscript, we have added some new results and discussion on this point in the revised manuscript and supplemental material (see Page 7, lines 182-184; Supplementary Fig. S6).

Figure R1 a. MFM image taken at $E = +3$ kV/cm under zero magnetic field for the $d \sim 350$ nm nano-dots of the $[\text{Pt}/\text{Co}/\text{Ta}]_{12}$ heterostructure. The distance between MFM tip and sample is fixed to be 30 nm. **b.** MFM image taken at $E = +3$ kV/cm with different tip-sample distance. **c.** MFM images taken at different tip-sample distance by reversing the direction of the magnetic field.

Referee C's Comment #2: lines 152-162: I find the description of the PMN-PT a bit incomplete. What is the orientation of the substrate? Is it a single crystal? Does the reversal occur by the domain nucleation and growth? If yes, what is the typical size of the domains compared to the discs or the skyrmions? Could it play a role?

Authors' Reply: We thank the reviewer for careful reading of our manuscript. In this work, we chose the (001)-oriented single-crystalline PMN-PT as the ferroelectric substrate of the ferromagnetic/ferroelectric heterostructure. We propose that the reversal of magnetic domain is closely related to the ferroelectric domain nucleation and growth. A ferroelectric is an insulating system that possesses two or more discrete stable or metastable states with non-zero electric polarization (P) in zero electrical field (E), referred to as "spontaneous polarization"^[1]. When a certain electrical field is applied, these polarized states can be modified through the coupling of the electric field to the polarization ($-E \cdot P$)^[1]. Moreover, the crystal structure of ferroelectric is closely coupled with the polarized states, which makes that the ferroelectric crystal is generally elongated along the polarized direction^[1]. On basis of the two important features, the crystal structure of the ferroelectric can be modified by the electric field. Together with such a structural modification, a strain can be further generated in the ferroelectric crystal^[1]. In the case of our experiments, as E increases from 0 to the polarized reversal field (+3 kV/cm) of PMN-PT substrate, a tensile strain is generated and its strength increases with the growth of the ferroelectric domain. Meanwhile, we find that with the

increase of tensile strain, the stripe gradually transforms into skyrmion (see Fig. 2 in the main text). As we further increase E above the polarized reversal field of +3 kV/cm, the ferroelectric domain is reversed and gradually grows with the increase of E . This feature makes the tensile strain gradually changes to a compressive type and increase with the increase of E . In contrast to the assisting effect of the tensile strain on the nucleation of skyrmion, the compressive strain promotes the skyrmion to gradually switch back to the stripe domain. Based on the results above, we can find that the variation of ferroelectric domain generates the strain which further leads to the reversal of the magnetic domain states. Thus, we propose that the reversal of magnetic domain is closely related to the ferroelectric domain nucleation and growth.

The size of the ferroelectric domains in the (001)-oriented single-crystallized PMN-PT has been systematically investigated in previous literature^[2]. It is found that the size of the ferroelectric domains in the (001)-oriented single-crystallized PMN-PT is approximately 10 μm ^[8]. Such a size range of ferroelectric domain is much larger than that of the nano-dots (the diameter is equal to 350 nm), which confirms that the domain nucleation and growth may play an important role in the magnetic domain evolution. In the revised manuscript, we have added more descriptions on PMN-PT in the revised manuscript (see Page 4, lines 118-119).

References:

- [1] Rabe, K. M., Dawber, M., Lichtensteiger, C., Ahn, C. H. & Triscone, J. M. *Modern Ferroelectrics* (Springer, 2007).
- [2] Ba, Y. et al. Spatially Resolved Electric-Field Manipulation of Magnetism for CoFeB Mesoscopic Discs on Ferroelectrics. *Adv. Funct. Mater.* **28**, 1706448 (2018).

Referee C's Comment #3: line 179: "after turning off E " is unclear. What means "reserved"? Do you mean "restored"?

Authors' Reply: We thank the reviewer for careful reading of our manuscript. We use the term "reserved" to declare that the domain structure returns back to the stripe state after E decreases to 0 kV/cm. In the revised manuscript, the term "reserved" is replaced by "restored" based on the reviewer's comments (see Page 6, line 177).

Referee C's Comment #4: lines 216-217: The remnant ferroelectric state induces several

magnetic states: skyrmions, stripe and vortex. This is the most interesting/surprising point of the article. To substantiate the discussion about the pinning, I would suggest performing small magnetic field cycles. If the magnetic states at ferroelectric remanance results from pinning, then a field cycle should bring the magnetic system to another state. It would be rather simple to verify it, and it would strengthen very much the discussion.

Authors' Reply: We thank the reviewer for the valuable suggestions, which are greatly helpful for us to improve the quality of our manuscript. Our experimental results demonstrate that the electrical-field-driven switching of different magnetic states is non-volatile. We propose that both the large remnant strain of FE substrate and the magnetic hysteretic effect originating from the geometrically pinning, result in the non-volatile feature. For the non-volatility of skyrmions, we propose that the magnetic hysteretic effect is the main reason. Our experimental results have demonstrated that, as E increases from 0 to +3 kV/cm, the tensile strain induces the stripe to gradually transform into skyrmions (see Fig. 3 in the main text). After E is decreased from +3 kV/cm to 0 kV/cm, the skyrmions can be retained (see Fig. 3 in the main text). Since the value of E for skyrmion generation (+3 kV/cm) is nearly the same with the polarization reversal electric field (+3 kV/cm) of the PMN-PT substrate, the remnant strain is nearly 0% when E is decreased from +3 kV/cm to 0 kV/cm (see Fig. R2a). Hence, the remnant strain of FE substrate is not the main reason for the non-volatility of skyrmions. Instead, we propose that the magnetic hysteretic effect originating from the geometrically pinning is the main reason for the non-volatile feature of the skyrmions. To prove this point, we first induce the stripe domains to completely transform into skyrmions by increasing magnetic field (see Fig. R2b), and subsequently the magnetic field is decreased to zero. As shown in Fig. Fig. R2b, the skyrmions can be restored at a zero magnetic field though their sharp appears to be elongated. This result clearly demonstrates that magnetic hysteretic effect results in the non-volatile feature of skyrmions. We have also carried out the same measurements in a continuous thin film (see Fig. R2c). However, few skyrmions are restored at zero magnetic field. This feature further suggests that the magnetic hysteretic effect in our experiments mainly originates from the geometrical pinning of the nano-dots, as reported in the previous literatures^[1-3].

In the case of the non-volatility of the vortex state, we propose that the large remnant strain of FE substrate plays a dominated role. Our experimental results have demonstrated that, as E increases from 0 to +10 kV/cm, the compressive strain induces the formation of the vortex state. This feature suggests that the compressive strain is the key factor for the stabilization of vortex. After E decrease from +10 kV/cm to 0 kV/cm, the vortex can be retained. As shown in Fig. 2b in the main text and Fig. R2a, when E decreases from +10 kV/cm to 0 kV/cm, a large remnant compressive strain of approximately 0.17% can be obtained as a result of the hysteretic characteristic of the PMN-PT substrate. However, if we further decrease the compressive strain slightly, the vortex gradually transforms into the skyrmion. This feature suggests that a large enough compressive strain is essential for the stabilization of vortex. On the other hand, we find that the vortex state at $E = 0$ kV/cm can reform in the nano-dot after it is destructed by applying an external out-of-plane magnetic field. Fig. R3 show a typical MFM image at $E = 0$ kV/cm after E decreases from +10 kV/cm. We can find that the nano-dots host vortex states at $E = 0$ kV/cm. By increasing the magnetic field from 0 mT to 50 mT, the vortex states gradually transform into the out-of-plane single domain. When the magnetic field is decreased to 0 mT, the vortex domains reform in the nano-dots. If the magnetic hysteretic is the key factor for the stabilization of vortex at $E = 0$ kV/cm, the out-of-plane single domain would transform into the stripe domain or skyrmions after decreasing the external magnetic field to zero (see Fig. R3b). This feature further confirms that the dominated factor for the stabilization of vortex at $E = 0$ kV/cm is not the magnetic hysteretic effect but the large remanent strain from the ferroelectric substrate. In the revised manuscript, we have added more discussions on the non-volatile feature (see Pages 8-9, lines 218-230, 234-243, Supplementary Figs. S12-13).

References:

- [1] Caretta, L. et al. Fast current-driven domain walls and small skyrmions in a compensated ferrimagnet. *Nat. Nanotech.* **13**, 1154-1160 (2018).
- [2] Hou, Z. et al. Manipulating the Topology of Nanoscale Skyrmion Bubbles by Spatially Geometric Confinement. *ACS Nano* **13**, 922 (2019).
- [3] Ho, P. et al. Geometrically Tailored Skyrmions at Zero Magnetic Field in Multilayered Nanostructures. *Phys. Rev. Appl.* **11**, 024064 (2019).

Figure R2 a. The remanent strain of $[\text{Pt}/\text{Co}/\text{Ta}]_8$ heterostructure for different electric field (E). The remanent strain is measured after decreasing E from a target value to zero. **b.** Magnetic field-dependent domain evolution process of the $[\text{Pt}/\text{Co}/\text{Ta}]_{12}$ heterostructure. **c.** Magnetic field-dependent domain evolution process of the $[\text{Pt}/\text{Co}/\text{Ta}]_{12}$ continuous thin film.

Figure R3 a. Magnetic field-dependent domain evolution process of the $[\text{Pt}/\text{Co}/\text{Ta}]_8$ heterostructure. These images are taken at $E = 0$ kV/cm after decreasing E from +10 kV/cm. **b.** Magnetic field-dependent domain evolution process with the magnetic field decreasing from 100 mT to 0 mT.

Referee C's Comment #5: Eq.1, 1.276: Please precise about the K_{eff} : it is the effective magnetic anisotropy of the domain wall or the film?

Authors' Reply: We thank the reviewer for the valuable comments. In our manuscript, the absolute value of domain wall surface energy density (σ_{DW}) can be calculated with the equation:

$$\sigma_{\text{DW}} = 4\sqrt{AK} - \pi|D|, \quad (1)$$

, where K is the magnetic anisotropy constant that accounts for the energy difference between the spin in the domain and that in the middle of the domain wall. Thus, K should represent the magnetic anisotropy of the domain wall. In this work, the value of K is proposed to be approximately equal to that of the thin film, as is described in previous reports^[1-3]. Below, we will discuss the approximation is reasonable.

For a (001)-oriented fcc Co thin film, the effective magnetic anisotropy energy of magnetic domain (E_{domain}) can be written as follow^[4]:

$$E_{domain} = K_{eff} \sin^2 \theta, \quad (2)$$

where K_{eff} is the effective magnetic anisotropy energy of the magnetic film and θ is the angle between the magnetization and normal direction of the film. If the magnetic domain possesses a 180° domain wall, the effective magnetic anisotropy energy of the domain wall (E_{wall}) can be written as follow^[5]:

$$E_{wall} = K_{eff} \sin^2 \theta - (9/4)(C_{11} - C_{12})\lambda_{001}^2 \left(\cos^2 \theta - \frac{1}{3} \right), \quad (3)$$

where C_{11} , C_{12} are the usual elastic moduli for a cubic crystal, and the λ_{001} is the magnetostriction constant of the (001) crystal plane. Since K in the equation (1) represents the energy difference between the spin in the domain and that in the middle of the domain wall, K can be further given by the equation:

$$K = \left| E_{domain}(\theta = 0) - E_{wall}(\theta = \frac{\pi}{2}) \right| = \left| K_{eff} + (3/4)(C_{11} - C_{12})\lambda_{100}^2 \right|. \quad (4)$$

For fcc-Co, the values of C_{11} , C_{12} , and λ_{100} are summarized in the Table R3. Thus, K can be further given by the equation:

$$K = \left| E_{domain}(\theta = 0) - E_{wall}(\theta = \frac{\pi}{2}) \right| = \left| K_{eff} + 301 J/m^3 \right|. \quad (5)$$

In our experiments, the value of K_{eff} is in the order of $10^5 J/m^3$ that is much larger than the difference between the values of K and K_{eff} (in the order of $10^2 J/m^3$). Thus, the approximation that value of K is equal to that of K_{eff} is reasonable. In the revised manuscript, this point has been declared (see Page 10 lines 282-285).

	C_{11} ($10^{11} J/m^3$)	C_{12} ($10^{11} J/m^3$)	λ_{100} (10^{-6})
Co (fcc)	2.42	1.60	70

Table R3 The values of elastic moduli for a cubic crystal^[6,7]

References:

- [1] Woo, S. et al. Observation of room-temperature magnetic skyrmions and their current-driven dynamics in ultrathin metallic ferromagnets. *Nat. Mater.* **15**, 501 (2016).
- [2] Pellegrin, J. P., Lau, D. & Sokalski, V. Dispersive Stiffness of Dzyaloshinskii Domain

Walls. *Phys. Rev. Lett.* **119**, 027203 (2017).

[3] Srivastava, T. et al. Mapping different skyrmion phases in double wedges of Ta/FeCoB/TaO_x trilayers. *Phys. Rev. B* **100**, 220401(R) (2019).

[4] Hellman, F. *Encyclopedia of Materials: Science and Technology* (Elsevier, Amsterdam, 2001).

[5] Baltzer, P. K. Effective magnetic anisotropy and magnetostriction of monocrystals. *Phys. Rev.* **108**, 580 (1957)

[6] Takahashi, H. et al. Measurement of magnetostriction constants in polycrystalline alloy and multilayer films of PdCo and PtCo. *J. Magn. Magn. Mater.* **126**, 282-284 (1993).

[7] Tian, Z., Sander, D. & Kirschner, J. Nonlinear magnetoelastic coupling of epitaxial layers of Fe, Co, and Ni on Ir (100). *Phys. Rev. B* **79**, 024432 (2009).

Referee C's Comment #6: page 10 and 11: The magnetic parameters are indicated with extremely high precision. Can the authors give error margin to all of them. They should also discuss what is the displayed error margin. For example, in Fig.4, the error margins are just “relative”, authors probably assumed a perfect knowledge of the saturation magnetization (M_s) or exchange energy (A). I doubt that they can determine M_s much better than within a few percent precision, or even few tens of % for A ...

Authors' Reply: We sincerely thank the reviewer for careful reading of our manuscript. The suggestions and comments are greatly helpful for us to improve the quality of our manuscript. In this work, we have experimentally established the E -dependent variation of the magnetic parameters, i.e., saturation magnetization (M_s), effective magnetic anisotropy (K_{eff}), magnetic exchange constant (A), and DMI constant (D), to explain the physical mechanism of the E -driven switching of different magnetic states. The E -dependent M_s and K_{eff} can be obtained by measuring the E -dependent magnetization curves of the heterostructure. We find that the variation of E affects M_s slightly and the value of M_s can be established to be 697 ± 7 kA/m³. The error margin of M_s is added by summarizing the values of M_s under different E . However, K_{eff} is significantly affected by the variation of E and the E -dependent value of K_{eff} is presented in Fig. 4a in the main text and Supplementary Table S2. The error margin of K_{eff} is established by measuring the two different samples. Since the E -dependent magnetization curves of the heterostructure can be well established, the errors of M_s and K_{eff} fall within a few percent precision.

To establish the value of A , we have measured the temperature-dependent saturation

magnetization $M_s(T)$, as shown in Fig. R4. The $M_s(T)$ curve can be fitted with the Bloch law^[1,2]:

$$M_s(T) = M_0(1 - b/T^{3/2}), \quad (1)$$

where M_0 is the saturation magnetization at zero Kelvin M_0 and b is a constant. The constant b can be further given by:

$$A = \frac{nk_B S^2}{a} \left(\frac{C}{b}\right)^{2/3}, \quad (2)$$

where $n = 4$ is the coordination number for a fcc lattice^[1,2] (the Co that deposited at room-temperature using sputter generally crystallizes into the fcc structure^[2,3]), k_B is the Boltzmann constant that takes a value of 1.380649×10^{-23} J/K, S is the spin per atom that takes the value of 1^[2,4], C is a constant that takes the value 0.0294 for an fcc lattice^[1,2], and a is the lattice constant of fcc Co that takes a value of 0.3536 nm^[2,5]. Thus, if the value of b is established, we can directly calculate the value of A by using equation (3). To determine the value of b , the temperature-dependent saturation magnetization of the [Pt/Co/Ta]₁₂ heterostructure at zero electrical field is fitted by using the Bloch law, as shown in Fig. R4. We find that the $M_s(T)$ curve agree well with the Bloch law over the temperature range of 60 K-330 K. However, when the temperature decreases below 60 K, the curve deviates from the Bloch law. Such deviations may be attributed to the pronounced role of the Ruderman-Kittel-Kasuya-Yosida (RKKY) interaction at the low-temperature range (below 60 K)^[6,7]. The corresponding value of A can be calculated to be 16.9 ± 0.2 pJ/m. The error bar is added by fitting the $M_s(T)$ curve within different temperature range and such a value of A falls within the reported range of A for the Co-based thin film^[4,7,8].

To investigate the effect of strain on the variation of A , we have further measured the temperature-dependent saturation magnetization of the [Pt/Co/Ta]₁₂ heterostructure at +10 kV/cm where the PMN-PT substrate exhibits a maximum strain. As shown in Fig. R4, we can find that the $M_s(T)$ curve keeps nearly unchanged at the whole temperature range of 10-330 K compared with that measured at a zero electrical field. Thus, the value of A should also be slightly affected under such a strain. That is to say, the value of A is insensitive to the strain in our experiments. On the other hand, previous experimental reports demonstrated that when a strain of 0.3% was applied on a FeGe thin plate, the value of T_c just varied by 3.3 K^[9]. Such a

slight variation of T_c only corresponds to a few percentage variation of A compared with that at the ambient pressure ($T_c = 280$ K)^[9]. For the [Pt/Co/Ta]₁₂ heterostructure in our experiments, the maximum tensile strain is 0.026% and the maximum compressive strain is 0.07%. These values are nearly one order of magnitude lower than that applied on the FeGe. Thus, we believe that the value of A is slightly affected by such a strain and it is reasonable for us to treat A as a strain-independent parameter in our experiments.

The value of σ_{DW} can be quantified by measuring the low-magnetic-field domain period (w) on the basis of the classical domain spacing model:

$$\frac{\sigma_{DW}}{\mu_0 M_s^2 t} = \frac{w^2}{t^2} \sum_{odd\ n=1}^{\infty} \left(\frac{1}{(\pi n)^3} \right) [1 - (1 - 2\pi n t/w) \exp(-2\pi n t/w)], \quad (3)$$

where t is the thickness of the film, M_s is the saturation magnetization, and w is the low-field domain period ($w = w_{\uparrow} + w_{\downarrow}$, w_{\uparrow} and w_{\downarrow} were the up and down domain widths, respectively) that can be obtained from the MFM data. The values of w are established directly measuring the period of the stripe domain on the experimental MFM images. Fig. R5 shows the typical contrast variation of the domain structures. We propose that the width between two neighbor contrast peaks is equal to period of a domain structure and their corresponding values at different electrical field (E) are summarized in Supplementary Table S2. Notably, the position of contrast peak is established by using a Gauss fitting and the error bar of the period is added by measuring the periods at different regions of a MFM image. On basis of counted values of w , the corresponding values of σ_{DW} at different electrical fields are summarized in Supplementary Table S2.

Having established the values of A , K_{eff} , and σ_{DW} , the value of D can be calculated with the equation:

$$\sigma_{DW} = 4\sqrt{AK_{eff}} - \pi|D|. \quad (4)$$

In our experiments, the values of A and M_s are proposed to be independent on the variation of E . Thus, the error bar of D under different E can be added on basis of the E -dependent error bars of K_{eff} and σ_{DW} . Since such E -dependent error bar corresponds for a fixed value of A and M_s , it can be called “relative error bar”, as pointed by the reviewer. In the revised manuscript, we have re-calculated the value of D on basis of the error margin of A (16.9 ± 0.2 pJ/m) and M_s

(697 ± 7 kA/m³), as shown in Fig. 4b. Moreover, we have added more discussions on the error margin of the magnetic parameters in the Methods section (see Page 13, lines 365-382).

References:

- [1] Chikazumi, S. *Physics of Ferromagnetism* (Oxford University Press, Oxford, 1997).
- [2] Shahbazi, K. et al. Domain-wall motion and interfacial Dzyaloshinskii-Moriya interactions in Pt/Co/Ir(t_{Ir})/Ta multilayers. *Phys. Rev. B* **99**, 094409 (2019).
- [3] Wells, A. W., Shepley, P. M., Marrows, C. H. & Moore, T. A. Effect of interfacial intermixing on the Dzyaloshinskii-Moriya interaction in Pt/Co/Pt. *Phys. Rev. B* **95**, 054428 (2017).
- [4] Balashov, T. et al. Magnetic anisotropy and magnetization dynamics of individual atoms and clusters of Fe and Co on Pt (111). *Phys. Rev. Lett.* **102**, 257203 (2009).
- [5] Liu, X. et al. Exchange stiffness, magnetization, and spin waves in cubic and hexagonal phases of cobalt. *Phys. Rev. B* **53**, 12166 (1996).
- [6] Tahiri, N., Ez-Zahraouy, H. & Benyoussef, A. Multilayer transition in a spin 3/2 Blume–Capel model with RKKY interaction. *Physica A* **388**, 3426-3432 (2009).
- [7] Knepper, J. W. & Yang, F. Y. Oscillatory interlayer coupling in Co/Pt multilayers with perpendicular anisotropy. *Phys. Rev. B* **71**, 224403 (2005).
- [8] Talapatra, A. & Mohanty, J. Laser induced local modification of magnetic domain in Co/Pt multilayer. *J. Magn. Magn. Mater.* **418**, 224-230 (2016).
- [9] Shibata, K. et al. Large anisotropic deformation of skyrmions in strained crystal. *Nature Nanotech.* **10**, 589 (2015).

Figure R4 Temperature-dependent saturation magnetization $M_s(T)$ of the [Pt/Co/Ta]₁₂ heterostructure at $E = 0$ kV/cm (orange dots) and +10 kV/cm (blue dots). The experimental dots are fitted by using the Bloch law (red line).

Figure R5 a. An amplification MFM image. In this image, the regions that represent the domain wall, up domain and down domain are clearly marked. **b.** The contrast variation extracted from the MFM image. The distance (L) between two neighboring contrast peaks is proposed to be a domain period (w). The experimental data is fitted by Gauss function (red line) to establish the position of the peak.

Referee C's Comment #7: Fig.1c: Why reversing the scale of d ? It is very confusing! Moreover, the choice of field at which skyrmions are nucleated is a bit arbitrary. Indeed from the pictures of Fig.S2, one can see that skyrmions appear at lower field than indicated, coexisting with worm domains. Authors should rather indicate a range of field at which the transition occurs.

Authors' Reply: We thank the reviewer for the valuable comments. In the revised manuscript, the scale of d is revised and more MFM images showing the magnetic transition process are added into the Fig. 1c in the main text (see Page 17, Fig. 1c).

Referee C's Comment #8: Fig.1 caption: line 501 – ' d ' not ' D '! The caption does not describe what represent the data or the line!! line 502: the colors do not correspond to the boxes.

Authors' Reply: We thank the reviewer for careful reading of our manuscript. In the revised manuscript, this point has been revised (see Page17, lines 538-542).

Referee C's Comment #9: Fig.2 caption: line 548 – there is no box.

Authors' Reply: We thank the reviewer for careful reading of our manuscript. In the revised manuscript, the box has been added (see Page18, Figure 2b).

Reviewers' comments:

Reviewer #1 (Remarks to the Author):

The authors of the manuscript 'Electric-field-driven Non-volatile Multi-state Switching of Individual Skyrmions in a Multiferroic Heterostructure' have provided answers to all my questions. They have also done simulations to understand how the strain evolves in discs of different diameters compared to the strain in a continuous film. This is an important aspect for this work, because only the electric field dependent strain in continuous films can be measured, but the authors want to understand the physics in discs. Indeed, the strain in the disc turns out to be rather inhomogeneous, the strain at the top of a 350 nm diameter disc (maybe 80-100 nm thick) amounts to only half of that at the bottom (see Fig. S4b) and actually the strain at the rim of the disc goes down even further, maybe to close to 20% only (see Fig. S4a). However, the authors do not take these new findings into account for the interpretation of their data. For the quantitative analysis (e.g. Fig. 2) they use the full strain for a continuous film instead of the reduced and inhomogeneous strain distribution according to their new simulations. When the edge plays such a fundamental role for the experimental observations (claimed to be responsible for the Skyrmion non-volatility, i.e. hysteresis of the magnetic state) then the role of the edge (e.g. modified strain) should be included in the analysis and interpretation.

Also, as far as I understand, the effective anisotropy plotted in Fig. 4a is derived from the continuous film (Fig. S15), but these values are then used to discuss the physics of discs. The discs are expected to exhibit a different anisotropy compared to the film because of the rim and because of the different strain, which in addition is inhomogeneous within the disc. Using these continuous film effective anisotropies (Fig. 4a) and the electric field dependent magnetic stripe domain period (Fig. S16) the authors derive a strain-dependence of D (Fig. 4b). The manuscript deals with the electric-field induced changes of the magnetic state, and I am surprised that the authors do not mention explicitly that the magnetic stripe domain structure in the film does not change when an electric field between 0 and 10kV/cm is applied (see Fig. S16 and the constant period in Tab. S2). When the magnetic state does not change either the parameters do not change or they exactly keep the balance; the latter option is the scenario the authors use, thus according to Eq. (1) a change in K (Fig. 4a) must then directly result in the according change of D (Fig. 4b) (because σ is the constant within the error and A is kept constant too). However, all this relies on K measured in a continuous film in which the magnetic state was shown to be independent on the electric field.

While I appreciate that the authors try to model and explain the underlying mechanisms that lead to their interesting observations in magnetic discs, I consider the model they use oversimplified, and I think that some physical insight might be missed. The limitations of the model are not discussed, instead the authors make a very strong claim that the effective anisotropy and the DM interaction are both changed by the strain in the film. While this might be the case, in my opinion this conclusion cannot be drawn rigorously. Therefore I cannot recommend publication in Nature Communications.

Reviewer #2 (Remarks to the Author):

The authors have addressed the comments and questions in a satisfactory way. They also have improved the manuscript.

Therefore, I recommend this paper to be published.

Reviewer #3 (Remarks to the Author):

The authors answered (lengthy but mostly correctly) all the remarks from all authors and applied related corrections to the manuscript. I think that the article is now ready for publication.

Response to the Report of Referee A

Referee A's Comment #1: The authors of the manuscript 'Electric-field-driven Non-volatile Multi-state Switching of Individual Skyrmions in a Multiferroic Heterostructure' have provided answers to all my questions. They have also done simulations to understand how the strain evolves in discs of different diameters compared to the strain in a continuous film. This is an important aspect for this work, because only the electric field dependent strain in continuous films can be measured, but the authors want to understand the physics in discs. Indeed, the strain in the disc turns out to be rather inhomogeneous, the strain at the top of a 350 nm diameter disc (maybe 80-100 nm thick) amounts to only half of that at the bottom (see Fig. S4b) and actually the strain at the rim of the disc goes down even further, maybe to close to 20% only (see Fig. S4a). However, the authors do not take these new findings into account for the interpretation of their data. For the quantitative analysis (e.g. Fig. 2) they use the full strain for a continuous film instead of the reduced and inhomogeneous strain distribution according to their new simulations. When the edge plays such a fundamental role for the experimental observations (claimed to be responsible for the Skyrmion non-volatility, i.e. hysteresis of the magnetic state) then the role of the edge (e.g. modified strain) should be included in the analysis and interpretation.

Authors' Reply: We thank the reviewer for careful reading of our manuscript. We also thank the reviewer for giving valuable comments. Following the reviewer's comments, we have discussed about the role of inhomogeneous strain in the analysis and interpretation parts of our revised manuscript.

First of all, we agree with the reviewer that it is misleading and inaccurate to use the full and homogeneous strain (ϵ) on the continuous thin film to describe the reduced and inhomogeneous ϵ on the nanodot. In the newly revised manuscript, we introduce the average strain (ϵ_{ave}), which represents an overall result of the inhomogeneous ϵ , to describe the electrical field (E)-dependent ϵ on the $d \sim 350$ nm nanodot. In order to calculate ϵ_{ave} , the corresponding ϵ distribution should be established. At current stage, however, it is really a challenge to experimentally map the ϵ distribution on such a small

nanodot. Instead, we derive the ε distribution by computational simulations. Fig. R1 shows a side view of the simulated ε distribution on the $d \sim 350$ nm nanodot that is subjected to a constant biaxial strain from the PMN-PT substrate. It is found that the value of ε decreases along both the radius (r) and thickness (t) directions. Such an inhomogeneous ε distribution can be further expressed as functions of both r and t . For $t \leq 20$ nm, the ε distribution is homogeneous and the value of ε is approximately equal to that of the substrate (ε_{sub}). For $t \geq 20$ nm, the relationship between ε , t , and r can be given as

$$\varepsilon(t, r) = \varepsilon_{\text{sub}} \exp[-7.8 \times 10^{-3}(t - 20) \exp(8.9 \times 10^{-3}r)], \quad (1)$$

where t ranges from 20 to 60 nm and r ranges from 0 to 175 nm. Based on the simulated ε distribution, ε_{ave} is calculated to be approximately 75% of the strain generated at the PMN-PT substrate (ε_{sub}) for the $d \sim 350$ nm [Pt/Co/Ta]₁₂ nanodot and 91% for the $d \sim 350$ nm [Pt/Co/Ta]₈ nanodot. In our experiments, the E - ε_{sub} curve can be directly measured by using a resistance-strain gauge that is glued to the top surface of the PMN-PT substrate (see Fig. R2). On basis of the E - ε_{sub} curve and the relationship between ε_{sub} and ε_{ave} , we can hence obtain the E - ε_{ave} curve of the $d \sim 350$ nm nanodot (see Fig. R3). In our newly revised manuscript, we replace the E - ε curves of the continuous thin film with the derived E - ε_{ave} curves of the nanodot (see in Fig. 2 in the main text) and we also add related descriptions into the main text (see Page 6, lines 155-163). Moreover, Fig. R1 and Fig. R2 are added into the Supplementary Information (see Fig. S3 and Fig. S4)

Next, we will discuss the relationship between inhomogeneous ε and K_{eff} on the $d \sim 350$ nm nanodot. As known to us, ε is closely coupled with K_{eff} ^[1,2]. Thus, the inhomogeneous ε distribution should lead to an inhomogeneous K_{eff} distribution. In this work, we use the average (K_{ave}) of the inhomogeneous K_{eff} to describe the experimentally measured K_{eff} on the $d \sim 350$ nm nanodot. Since K_{ave} corresponds to ε_{ave} , we replace the K_{eff} - ε curve shown in Fig. 4a in the last version of our manuscript with the K_{ave} - ε_{ave} curve in the newly revised one. On the other hand, for a magnetic system subjected to a biaxial in-plane ε , the relationship between K_{ave} and ε_{ave} can be given as^[1,2]

$$K_{\text{ave}} = K_0 + \Delta K_{\text{ave}} = K_0 - \frac{3}{2} Y \lambda_s \frac{2\nu}{1-\nu} \varepsilon_{\text{ave}}, \quad (2)$$

where K_0 is the initial magnetic anisotropy without strain, Y is the Young's moduli, λ_s is

the saturation magnetostriction constant, and ν is the Poisson's constant. Equation (2) clearly demonstrates that K_{ave} is linearly dependent on ε_{ave} . In the case of our experiments, the experimentally established ε_{ave} -dependent K_{ave} can be well fitted by using such a linear equation (see Fig. R4a) and the equation can be given as

$$K_{\text{ave}} = 5.1 \times 10^6 \varepsilon_{\text{ave}} + K_0, \quad (3)$$

where K_0 is the effective magnetic anisotropy of $d \sim 350$ nm nanodot at unstrained state and is equal to $(-0.90 \pm 0.02) \times 10^5$ J/m³.

The corresponding absolute value of D_{ave} could be calculated with the equation

$$\sigma_{\text{DW}} = 4\sqrt{AK_{\text{ave}}} - \pi|D_{\text{ave}}|, \quad (4)$$

where σ_{DW} is the domain wall surface energy density and A is the exchange constant. As ε_{ave} -dependent K_{ave} has been obtained above, the relationship between ε_{ave} and σ_{DW} should be experimentally established to calculate ε_{ave} -dependent D_{ave} . The value of σ_{DW} can be quantified by measuring the low-magnetic-field domain period (w) on the basis of a domain spacing model

$$\frac{\sigma_{\text{DW}}}{\mu_0 M_s^2 t} = \frac{w^2}{t^2} \sum_{\text{odd } n=1}^{\infty} \left(\frac{1}{(\pi n)^3} \right) [1 - (1 - 2\pi n t/w) \exp(-2\pi n t/w)], \quad (5)$$

where t is the thickness of the film, M_s is saturation magnetization, and w is the low-field domain period ($w = w_{\uparrow} + w_{\downarrow}$, w_{\uparrow} and w_{\downarrow} were the up and down domain widths, respectively) that can be obtained from the MFM data. For the continuous thin film or nanodot with a relatively large diameter ($d \geq 850$ nm), they host periodic magnetic stripe domains. However, when d is smaller than 850 nm ($d < 850$ nm), the strong geometrical confinement induces the periodic stripe domain to transform into the non-periodic one (see Fig. R4b). Thus, we can no longer directly calculate D_{ave} of the $d \sim 350$ nm nanodot based on K_{ave} and σ_{DW} . Instead, we derive it on basis of the D_{ave} - ε_{ave} relationship established on the continuous thin film and $d \sim 850$ nm nanodot. In Fig. R5a, we show a series of D_{ave} derived at different ε_{ave} on the continuous thin film. We can find that the absolute value of D_{ave} decreases with increasing tensile strain ($\varepsilon_{\text{ave}} < 0$) while increases with increasing compressive strain ($\varepsilon_{\text{ave}} > 0$). Such a change tendency of D_{ave} with ε_{ave} is consistent with that reported in recent literatures^[3-7], which thus validates our results. We have also derived the ε_{ave} -dependent D_{ave} on the $d \sim 850$ nm nanodot, as shown in Fig. R5a.

The corresponding change tendency of D_{ave} with ε_{ave} is similar to that on the continuous thin film, which confirms that D_{ave} is closely coupled with ε_{ave} . Moreover, we find that the ε_{ave} -dependent D_{ave} on the $d \sim 850$ nm nanodot nearly overlaps that on the continuous thin film though the ε_{ave} range of the continuous thin film is larger than that of the $d \sim 850$ nm nanodot. This feature further suggests that the relationship between D_{ave} and ε_{ave} is little affected by the geometrically confined effect, namely, the relationship between D_{ave} and ε_{ave} is approximately fixed no matter on the continuous thin film or the nanodot. That can be attributed to the intrinsic feature of D . As known to us, D directly originates from the spin-orbital coupling effect of the film interface. Since the spin-orbital coupling effect depends on the electronic structure of the film, it can be little affected by the geometrical confinement. However, the strain may show a significant influence on D because it can affect the electronic structure and spin-orbital coupling effect of the film by tuning its lattice parameter^[8-10]. Thus, we have observed that the value of D can be changed by strain and the relationship between D and strain is little affected by the geometrical confinement. To evaluate ε_{ave} -dependent D_{ave} on the $d \sim 350$ nm nanodot, the relationship between D_{ave} and ε_{ave} should be quantitatively established. In recent reports^[6,7], D_{ave} is proposed to vary with ε_{ave} approximately in a linear manner. In our work, a linear equation can also be derived by fitting the ε_{ave} -dependent D_{ave} for both the continuous thin film and $d \sim 850$ nm nanodot (see Fig. R5a). Based on the fitted ε_{ave} - D_{ave} equation and the ε_{ave} on the $d \sim 350$ nm nanodot, we can hence derive the values of D_{ave} on the $d \sim 350$ nm nanodot at different ε_{ave} , as shown in Fig. R5b. The error bar is added by fitting different ε_{ave} - D_{ave} equations. In the newly revised manuscript, we add this section into the Supplementary Information (see Fig. S17 and S17 Note).

Finally, we will discuss the influence of inhomogeneous $K_{\text{eff}}&D$ distribution on the simulated domain evolution processes. In the micromagnetic simulations shown in Fig. R5, the $K_{\text{eff}}&D$ distribution is set to be inhomogeneous based on the simulated ε distribution and the experimentally established relationship between $K_{\text{eff}}&D$ and ε (see Fig. R6). Details about the simulations are presented in the Fig. R6 Note. We find that the change tendency of the magnetic states with the inhomogeneous $K_{\text{eff}}&D$ is similar to that with the homogeneous one. This result suggests that the inhomogeneous feature of $K_{\text{eff}}&D$ does not

play a crucial role in the strain-induced domain transition process of the nanodot. In the newly revised manuscript, we replace Figs. 4c and 4d with the Figs. R6a and 6b to make the simulations to agree the experiments better. Meanwhile, Fig. R6 Note is added into Supplementary Information to describe the simulation process in detail (see Supplementary Note for micromagnetic simulations).

References:

- [1] Shepley, P., Rushforth, A., Wang, M. et al. Modification of perpendicular magnetic anisotropy and domain wall velocity in Pt/Co/Pt by voltage-induced strain. *Sci. Rep.* **5**, 7921 (2015).
- [2] Manchanda, P., Singh, U., Adenwalla, S., Kashyap, A. & Skomski, R. Strain and Stress in Magnetoelastic Co–Pt Multilayers. *IEEE Transactions on Magnetics* **50**, 1-4 (2014).
- [3] Shibata, K. et al. Large anisotropic deformation of skyrmions in strained crystal. *Nat. Nanotech.* **10**, 589-592 (2015).
- [4] Koretsune, T., Nagaosa, N. & Arita, R. Control of Dzyaloshinskii-Moriya interaction in $Mn_{1-x}Fe_xGe$: a first-principles study. *Sci. Rep.* **5**, 13302 (2015).
- [5] Kitchaev, D. A., Beyerlein, I. J. & Van, der. Ven. A. Phenomenology of chiral Dzyaloshinskii-Moriya interactions in strained materials. *Phys. Rev. B* **98**, 214414 (2018).
- [6] Zhang, W., Jiang, B., Wang, L., Fan, Y., Zhang, Y., et al. Enhancement of Interfacial Dzyaloshinskii-Moriya Interaction: A Comprehensive Investigation of Magnetic Dynamics. *Physical Review Applied*, **12**, 064031 (2019).
- [7] Gusev, N. S., Sadovnikov, A. V., Nikitov, S. A., Sapozhnikov, M. V. & Udalov, O. G. Manipulation of the Dzyaloshinskii–Moriya Interaction in Co/Pt Multilayers with Strain. *Phys. Rev. Lett.* **124**, 157202 (2020).
- [8] Koretsune, T., Nagaosa, N. & Arita, R. Control of Dzyaloshinskii-Moriya interaction in $Mn_{1-x}Fe_xGe$: a firstprinciples study. *Sci. Rep.* **5**, 13302 (2015).
- [9] Kautzsch, L., et al. Controlling Dzyaloshinskii-Moriya interactions in the skyrmion host candidates $FePd_{1-x}Pt_xMo_3N$. *Phys. Rev. Materials* **4**, 024412 (2020).
- [10] Grytsiuk, S., et al. Ab initio analysis of magnetic properties of the prototype B20 chiral magnet FeGe. *Phys. Rev. B* **100**, 214406 (2019).

Figure R1 Side view of the simulated ε distribution on a $d \sim 350\text{nm}$ nanodot that is

subjected to a constant biaxial compressive strain of 0.4% from the PMN-PT substrate. The distribution of ε shows a symmetric feature and its value decreases along both the radius (r) direction and thickness (t) of the nanodot. The color represents the value of ε at this region.

Figure R2 **a** Schematic diagram for measuring the in-plane strain on the PMN-PT substrate. Details about the measurement processes are presented in Supplementary Fig. S3. **b** The measured E - ε_{sub} curve of the PMN-PT substrate.

Figure R3 Electric-field-induced binary and multi-state switching of individual skyrmion. The average strain ε_{ave} and corresponding magnetic domain evolution processes in the $d \sim 350$ nm **a** $[\text{Pt}/\text{Co}/\text{Ta}]_{12}$ and **b** $[\text{Pt}/\text{Co}/\text{Ta}]_8$ nano-dots in a cycle of E ranging from +10 to -10 kV/cm. Positive ε_{ave} (red dots) represents a tensile strain while negative ε_{ave} (blue dots) represents a compressive strain. $\mu_0 H$ represents the external magnetic field except that from the MFM tip and here $\mu_0 H$ is equal to be 0 mT. The inset of **b** illustrates the spin texture of the magnetic domain that is encompassed by the red boxes. The stripe domain enclosed by the black box shows the initial state of the magnetic domain evolution path. The gray dots represent the corresponding electric field for the MFM images. The MFM contrast represents the MFM tip resonant frequency shift (Δf).

Figure R4 Dependence of the experimentally established values of K_{ave} on ϵ_{ave} on the $d \sim 350$ nm nanodot. The positive value of ϵ_{ave} represents the tensile strain, while the negative value signifies the compressive strain. The data is fitted by using the linear equations (black line). **b** E -dependent MFM images of continuous thin film, $d \sim 850$ nm nanodots, $d \sim 600$ nm of nanodots and $d \sim 350$ nm nanodots, without external magnetic field. The scale bar is 500 nm.

Figure R5. a Dependence of the experimentally established D_{ave} on ϵ_{ave} for the continuous thin film (red circle) and $d \sim 850$ nm nanodot (blue square). The black line represents the fitted equation for the data points of both continuous thin film and $d \sim 850$ nm nanodot. **b** Dependence of the evaluated D_{ave} on ϵ_{ave} for $d \sim 350$ nm nanodot. The black line represents the fitted equation for the data.

Figure R6 The influence of **a** D_{ave} and **b** K_{ave} on the magnetic domain evolution. Notably, when one magnetic parameter varies in the simulations, the other magnetic parameters are fixed. An external magnetic field of 100 mT is applied in the simulations. The magnetic domain enclosed by the black dashed boxes in **a** and **b** represent the initial states in the domain evolution process.

Figure R6 Note: Based on the simulations (see Fig. R1), the ε distribution on the $d \sim 350$ nm nanodot can be expressed as:

$$\varepsilon(t, r) = \begin{cases} 1.3\varepsilon_{ave} & 0\text{nm} \leq t \leq 20\text{nm} \\ 1.3\varepsilon_{ave} \exp[-7.8 \times 10^{-3}(t - 20) \exp(8.9 \times 10^{-3}r)] & 20\text{nm} < t \leq 60\text{nm} \end{cases} \quad (6)$$

,where ε_{ave} is the average strain of the nanodot, t is the thickness of the nanodot (ranging from 0 to 60 nm), and r is the radius of the nanodot (ranging from 0 to 175 nm). By fitting the ε_{ave} - K_{ave} and ε_{ave} - D_{ave} curves in Fig. R4a and Fig. R5b, the ε_{ave} -dependent K_{ave} and D_{ave} on the nanodot can be expressed as:

$$K_{ave} = 5.1 \times 10^6 \varepsilon_{ave} + K_0 \quad (7)$$

$$D_{ave} = -3.6 \times 10^6 \varepsilon_{ave} + D_0 \quad (8)$$

,where K_0/D_0 is effective magnetic anisotropy/DMI constant at the unstrained state and K_{ave}/D_{ave} is the average K_{eff}/D of the nanodot. By combining the equations (7) and (8) with equation (6), the K_{eff} & D distribution on the nanodot can be expressed as

$$K_{eff}(t, r) = \begin{cases} 1.3(K_{ave} - K_0) + K_0 & 0\text{nm} \leq t \leq 20\text{nm} \\ 1.3(K_{ave} - K_0) \exp[-7.8 \times 10^{-3}(t - 20) \exp(8.9 \times 10^{-3}r)] + K_0 & 20\text{nm} < t \leq 60\text{nm} \end{cases} \quad (9)$$

$$D(t, r) = \begin{cases} 1.3(D_{ave} - D_0) + D_0 & 0\text{nm} \leq t \leq 20\text{nm} \\ 1.3(D_{ave} - D_0) \exp[-7.8 \times 10^{-3}(t - 20) \exp(8.9 \times 10^{-3}r)] + D_0 & 20\text{nm} < t \leq 60\text{nm} \end{cases} \quad (10)$$

To reduce the computation load of the simulations, we have simplified the model. Along the thickness direction, the model is divided into three layers (each layer is 20 nm in thickness). For the bottom layer ($t \leq 20$ nm), since the K_{eff}/D distribution is homogeneous, it is no need to be simplified. In the case of the middle layer ($20 \text{ nm} < t \leq 40 \text{ nm}$) and top layer ($40 \text{ nm} < t \leq 60 \text{ nm}$), we propose that the K_{eff}/D distribution along the thickness is homogenous for a fixed r but varies linearly with r . Then, the equations (9) and (10) can be simplified as:

$$K_{\text{eff}}(t, r) = \begin{cases} 1.3(K_{\text{ave}} - K_0) + K_0 & 0\text{nm} \leq t \leq 20\text{nm} \\ 1.3(K_{\text{ave}} - K_0)(0.9 - 4.6 \times 10^{-4}r) + K_0 & 20\text{nm} < t \leq 40\text{nm} \\ 1.3(K_{\text{ave}} - K_0)(0.6 - 1.4 \times 10^{-3}r) + K_0 & 40\text{nm} < t \leq 60\text{nm} \end{cases} \quad (11)$$

$$D(t, r) = \begin{cases} 1.3(D_{\text{ave}} - D_0) + D_0 & 0\text{nm} \leq t \leq 20\text{nm} \\ 1.3(D_{\text{ave}} - D_0)(0.9 - 4.6 \times 10^{-4}r) + D_0 & 20\text{nm} < t \leq 40\text{nm} \\ 1.3(D_{\text{ave}} - D_0)(0.6 - 1.4 \times 10^{-3}r) + D_0 & 40\text{nm} < t \leq 60\text{nm} \end{cases} \quad (12)$$

Having established the relationship between K_{eff}/D distribution and $K_{\text{ave}}/D_{\text{ave}}$ on the $d \sim 350$ nm nanodot, we hereafter simulated the influence of the variation of $K_{\text{ave}}/D_{\text{ave}}$ on the magnetic domain evolution process. The dynamics of magnetization are governed by the Landau-Lifshitz-Gilbert (LLG) equation

$$\frac{dm}{dt} = -\gamma_0 m \times h_{\text{eff}} + \alpha \left(m \times \frac{dm}{dt} \right), \quad (13)$$

where t is the time, γ_0 is the gyromagnetic ratio with absolute value, and α is the Gilbert damping coefficient. In addition, $m = M/M_s$ is the reduced magnetization with saturation magnetization M_s , $h_{\text{eff}} = -\partial E_{\text{den}}/\partial m$ is the effective field, and the energy density E_{den} as a function of m includes the exchange energy, anisotropy energy, demagnetization energy, and Dzyaloshinskii-Moriya energy. Since the magnetic parameters, e.g., M_s and A , are insensitive to the strain, they are set to be homogeneous in our simulations ($M_s = 6.0 \times 10^5$ A/m, $A = 1.7 \times 10^{-11}$ J/m). The variation range of D_{ave} and K_{ave} is set to be $0.10 \sim 2.50$ mJ/m² and $0.00 \sim -2.50 \times 10^5$ J/m³, respectively. Furthermore, the Gilbert damping coefficient is 0.3. The simulated nano-dot has a diameter of 350 nm and thickness of 60 nm, and the mesh size was $5 \times 5 \times 20$ nm. The micromagnetic simulations are performed with the software package OOMMF. Notable, the simulation is performed at zero temperature and no thermal field is introduced.

Referee A's Comment #2: Also, as far as I understand, the effective anisotropy plotted in Fig. 4a is derived from the continuous film (Fig. S15), but these values are then used to discuss the physics of discs. The discs are expected to exhibit a different anisotropy compared to the film because of the rim and because of the different strain, which in addition is inhomogeneous within the disc.

Authors' Reply: We thank the reviewer for giving valuable comments, which are great helpful for the improvement of our manuscript. In fact, the effective anisotropy (K_{ave}) plotted in Fig. 4a in the last version of our manuscript is derived from the $d \sim 350$ nm nanodot, while the values of K_{ave} for the continuous thin film are presented in Table S2 to calculate the corresponding values of D . In our newly revised manuscript, we have revised related descriptions in the main text to clarify this point more clearly to avoid misleading readers (see page 10, lines 273-278 in the main text).

We agree with the reviewer that the $d \sim 350$ nm nanodot shows a different K_{ave}

compared with that of the thin film due to the effect of geometrical confinement. To illustrate their differences in detail, we have plotted their corresponding $K_{\text{ave}}-\varepsilon_{\text{ave}}$ curves in Fig. R7. For the continuous thin film, K_{eff} has a negative value of $(-1.10\pm 0.02)\times 10^5 \text{ J/m}^3$ at unstrained state. However, for the $d \sim 350 \text{ nm}$ nanodot, K_{ave} increases to $(-0.90\pm 0.02)\times 10^5 \text{ J/m}^3$. We can attribute the increase of K_{ave} to the increase of the shape anisotropy (K_s) originating from the reduction of sample lateral dimension.

Next, we will discuss the differences of K_{ave} at the strained state. For a magnetic system subjected to a biaxial in-plane strain, the relationship between K_{ave} and ε_{ave} can be given as^[1,2]:

$$K_{\text{ave}} = K_0 + \Delta K_{\text{ave}} = K_0 - \frac{3}{2}Y\lambda_s \frac{2\nu}{1-\nu} \varepsilon_{\text{ave}}, \quad (14)$$

where K_0 is the initial magnetic anisotropy at unstrained state, ΔK_{ave} is variation range of K_{ave} with ε_{ave} and equal to $-\frac{3}{2}Y\lambda_s \frac{2\nu}{1-\nu} \varepsilon_{\text{ave}}$, Y is the Young's moduli, λ_s is the saturation magnetostriction constant, and ν is the Poisson's constant. Equation (14) clearly demonstrates that K_{ave} is linearly dependent on ε_{ave} . Therefore, the dependence of K_{ave} on ε_{ave} (no matter for continuous thin film or nanodot) can be well fitted by using the linear equation, as shown in Fig. R7. Moreover, we find that the slope of equation (14) is a constant which is not related to the geometric dimension of the sample but only determined by the intrinsic parameters, i.e., Y , λ_s , ν . Therefore, the $K_{\text{ave}}-\varepsilon_{\text{ave}}$ curves of the continuous thin film and nanodot show a similar slope (see Fig. R7). Based on the discussions above, we can conclude that the $d \sim 350 \text{ nm}$ nanodot shows a different K_{ave} compared with that of the thin film while their corresponding change tendency of K_{ave} with ε_{ave} is nearly the same.

References:

- [1] Shepley, P., Rushforth, A., Wang, M. et al. Modification of perpendicular magnetic anisotropy and domain wall velocity in Pt/Co/Pt by voltage-induced strain. *Sci. Rep.* **5**, 7921 (2015).
- [2] Manchanda, P., Singh, U., Adenwalla, S., Kashyap, A. & Skomski, R. Strain and Stress in Magnetoelastic Co–Pt Multilayers. *IEEE Transactions on Magnetics* **50**, 1-4 (2014).

Figure R7 Dependence of the experimentally established K_{ave} on ϵ_{ave} for the continuous thin film (red circle) and $d \sim 350$ nm nanodot (red square). The ϵ_{ave} -dependent K_{ave} is fitted by using the linear equation (red line).

Referee A's Comment #3: Using these continuous film effective anisotropies (Fig. 4a) and the electric field dependent magnetic stripe domain period (Fig. S16) the authors derive a strain-dependence of D (Fig. 4b). The manuscript deals with the electric-field induced changes of the magnetic state, and I am surprised that the authors do not mention explicitly that the magnetic stripe domain structure in the film does not change when an electric field between 0 and 10kV/cm is applied (see Fig. S16 and the constant period in Tab. S2). When the magnetic state does not change either the parameters do not change or they exactly keep the balance; the latter option is the scenario the authors use, thus according to Eq. (1) a change in K (Fig. 4a) must then directly result in the according change of D (Fig. 4b) (because sigma is the constant within the error and A is kept constant too). However, all this relies on K measured in a continuous film in which the magnetic state was shown to be independent on the electric field.

Authors' Reply: We sincerely thank the reviewer for careful reading of our manuscript. The valuable suggestions and comments are greatly helpful to improve our manuscript. Following the reviewer's comments, we have re-checked the electric-field (E) induced changes of the stripe domain. Fig. R8a shows a series of MFM images taken at different E and different regions of the continuous thin film. The period (w) of the stripe domains recorded by these images is summarized in Fig. R8b. One can clearly find that, the value of w just shows a slight change with E , which confirms that our previous observations are

reliable. Based on the E -dependent w , we have further calculated the corresponding domain wall surface energy density (σ_{DW}), as shown in Fig. R8c. Since σ_{DW} is positively correlated with w , the value of σ_{DW} also varies slightly with that of E . As shown in equation (4) above, σ_{DW} is directly coupled with D and K_{eff} . In our experiments, we have confirmed that the value of K_{eff} varies significantly with that of E on the continuous thin film (see Fig. R7). If D is assumed to be fixed, the variation K_{eff} should lead to a significant change of σ_{DW} (see Fig. R9). Details about establishing the change of σ_{DW} induced by the variation of K_{eff} are presented in the caption of Fig. R9. Since σ_{DW} is positively correlated with w , a significant change of w for the stripe domains should also be observed in the MFM images, as is the case reported by Dai *et al*^[1]. However, our results demonstrate that w only changes slightly with E , as shown in Figs. 8a and 8b. Such a discrepancy suggests that D should also be changed by E (or ε) to offset the influence from the variation of K_{eff} on w . Therefore, we propose that both K_{eff} and D are changed by E (or ε) on the continuous thin film and an interplay of the two parameters results in a delicate balance of the stripe domain. We should note here that the E -dependent w , σ_{DW} , K_{eff} and D of the $d \sim 850$ nm nanodot have also been experimentally established (see Fig. R10). Their change tendency with E (or ε) is similar to that of the continuous thin film, which suggests that D of the $d \sim 850$ nm nanodot is also changed by E (or ε). In the newly revised manuscript, we add the related descriptions and discussions about the balance of stripe domains at different E into the Supplementary Information (see Table S2 Note).

For the $d \sim 350$ nm nanodot, as mentioned above, we can no longer directly calculate the corresponding D based on equation (4) because the strong geometrical confinement induces the periodic stripe domain to transform into the non-periodic one (see Fig. R4b). Instead, we can derive it on basis of the D - ε relationship established on the continuous thin film and $d \sim 850$ nm nanodot (see Fig. R5) because the relationship between D and ε is intrinsic and little affected by the geometrical confinement (detailed discussions are presented in Response to Referee A's Comment #1). Fig. R5b shows the derived ε -dependent D on the $d \sim 350$ nm nanodot. We can clearly find that the absolute value of D decreases with increasing tensile strain while increases with increasing compressive strain. Such a change tendency of D with ε is consistent with that reported in recent literatures^[2-6],

which thus confirms that our results are reliable. Based on the discussions above, we can conclude that both K_{eff} and D are changed by E (or ε) on the continuous thin film and nanodots ($d \sim 350$ nm, 850 nm) and an interplay of the two parameters results in a delicate balance of the stripe domain on the continuous thin film and $d \sim 850$ nm nanodot.

References:

- [1] Dai, G. et al. Stress tunable magnetic stripe domains in flexible $\text{Fe}_{81}\text{Ga}_{19}$ films. *J. Phys. D: Appl. Phys.* **53**, 055001 (2020).
- [2] Shibata, K. et al. Large anisotropic deformation of skyrmions in strained crystal. *Nat. Nanotech.* **10**, 589-592 (2015).
- [3] Koretsune, T., Nagaosa, N. & Arita, R. Control of Dzyaloshinskii-Moriya interaction in $\text{Mn}_{1-x}\text{Fe}_x\text{Ge}$: a first-principles study. *Sci. Rep.* **5**, 13302 (2015).
- [4] Kitchaev, D. A., Beyerlein, I. J. & Van, der. Ven. A. Phenomenology of chiral Dzyaloshinskii-Moriya interactions in strained materials. *Phys. Rev. B* **98**, 214414 (2018).
- [5] Zhang, W., Jiang, B., Wang, L., Fan, Y., Zhang, Y., et al. Enhancement of Interfacial Dzyaloshinskii-Moriya Interaction: A Comprehensive Investigation of Magnetic Dynamics. *Physical Review Applied*, **12**, 064031 (2019).
- [6] Gusev, N. S., Sadovnikov, A. V., Nikitov, S. A., Sapozhnikov, M. V. & Udalov, O. G. Manipulation of the Dzyaloshinskii–Moriya Interaction in Co/Pt Multilayers with Strain. *Phys. Rev. Lett.* **124**, 157202 (2020).

Figure R8 **a.** E -dependent MFM images of continuous thin film without external magnetic field. “#1”, “#2” and “#3” represent three different regions of the continuous thin film. The scale bar is 500 nm. **b.** E -dependent w on the continuous thin film. **c** E -dependent σ_{DW} on the continuous thin film.

Figure R9 E -dependent $\Delta\sigma_{\text{DW}}$ of the continuous thin film. $\Delta\sigma_{\text{DW}}$ represents the variation of σ_{DW} and is equal to $\sigma_{\text{DW}}(E) - \sigma_{\text{DW}}(0)$. Black dotted line represents E -dependent $\Delta\sigma_{\text{DW}}$ that is derived directly based on the E -dependent w presented in “#1” of Fig. R8b. The blue dotted line represents the $\Delta\sigma_{\text{DW}}$ that is derived based on σ_{DW} calculated by using K and D : $\sigma_{\text{DW}} = 4\sqrt{AK} - \pi|D|$. To demonstrate the influence of K on $\Delta\sigma_{\text{DW}}$, the value of D is fixed to be D_0 that is equal to the value of D on the continuous thin film at $E = 0$ kV/cm. Since E -dependent K is well established in our experiments (see Fig. R7), the values of σ_{DW} at different K can be calculated. $\Delta\sigma_{\text{DW}}$ is equal to the difference of $\sigma_{\text{DW}}(K)$ and $\sigma_{\text{DW}}(K_0)$, namely, $\Delta\sigma_{\text{DW}} = \sigma_{\text{DW}}(K) - \sigma_{\text{DW}}(K_0)$.

Figure R10 a. E -dependent MFM images of $d \sim 850$ nm nanodots without external magnetic field. “#1”, “#2” and “#3” represent three different nanodots. **b.** E -dependent w of the three nanodots marked in **a**. **c** E -dependent σ_{DW} calculated based on the values of w shown in **b**. **d** Dependence of the experimentally established values of K on ε_{ave} for the $d \sim 800$ nm nanodot. **e** Dependence of the derived values of D on ε_{ave} for the $d \sim 850$ nm nanodot.

Referee A’s Comment #4: While I appreciate that the authors try to model and explain the underlying mechanisms that lead to their interesting observations in magnetic discs, I consider the model they use oversimplified, and I think that some physical insight might be

missed. The limitations of the model are not discussed, instead the authors make a very strong claim that the effective anisotropy and the DM interaction are both changed by the strain in the film. While this might be the case, in my opinion this conclusion cannot be drawn rigorously. Therefore I cannot recommend publication in Nature Communications.

Authors' Reply: We thank the reviewer for giving valuable comments, which are great helpful for the improvement of our manuscript. Following the reviewer's comments and suggestions, we have added more insightful discussions about the role of inhomogeneous strain in the analysis and interpretation parts of our newly revised manuscript. First of all, we introduce the average strain (ε_{ave}), which represents an overall result of the inhomogeneous ε , to describe the electrical field (E)-dependent ε on the $d \sim 350$ nm nanodot. Moreover, the relationship between ε_{ave} and $K_{ave}\&D_{ave}$ on the $d \sim 350$ nm nanodot is established. Finally, the $K_{eff}\&D$ distribution is set to be inhomogeneous based on the simulated ε distribution and the experimentally established relationship between $K_{eff}\&D$ and ε , and the influence of inhomogeneous $K_{eff}\&D$ distribution on the simulated domain evolution processes has been discussed. We hope the reviewer will be satisfied with the revised manuscript and our responses.

Response to the Report of Referee B

Referee B's Comment: The authors have addressed the comments and questions in a satisfactory way. They also have improved the manuscript. Therefore, I recommend this paper to be published.

Authors' Reply: We sincerely thank the reviewer for recommending our manuscript to be published in Nature Communications.

Response to the Report of Referee C

Referee C's Comment: The authors answered (lengthy but mostly correctly) all the remarks from all authors and applied related corrections to the manuscript. I think that the article is now ready for publication.

Authors' Reply: We sincerely thank the reviewer for recommending our manuscript to be published in Nature Communications.